# Specificities of exosome versus small ectosome secretion revealed by live intracellular tracking of CD63 and CD9

Mathilde Mathieu [1,2], Nathalie Névo [1], Mabel Jouve[3], José Ignacio Valenzuela[4], Mathieu Maurin[1], Frederik J. Verweij[5], Roberta Palmulli [2,4], Danielle Lankar[1], Florent Dingli [6], Damarys Loew [6], Eric Rubinstein [7], Gaëlle Boncompain [4], Franck Perez [4] & Clotilde Théry [1✉]

Despite their roles in intercellular communications, the different populations of extracellular vesicles (EVs) and their secretion mechanisms are not fully characterized: how and to what extent EVs form as intraluminal vesicles of endocytic compartments (exosomes), or at the plasma membrane (PM) (ectosomes) remains unclear. Here we follow intracellular trafficking of the EV markers CD9 and CD63 from the endoplasmic reticulum to their residency compartment, respectively PM and late endosomes. We observe transient co-localization at both places, before they finally segregate. CD9 and a mutant CD63 stabilized at the PM are more abundantly released in EVs than CD63. Thus, in HeLa cells, ectosomes are more prominent than exosomes. By comparative proteomic analysis and differential response to neutralization of endosomal pH, we identify a few surface proteins likely specific of either exosomes (LAMP1) or ectosomes (BSG, SLC3A2). Our work sets the path for molecular and functional discrimination of exosomes and small ectosomes in any cell type.

[1] INSERM U932, Institut Curie Centre de Recherche, PSL Research University, Paris, France. [2] Université de Paris, Paris, France. [3] CNRS UMR3215, Institut Curie, PSL Research University, Paris, France. [4] CNRS UMR144, Institut Curie, PSL Research University, Paris, France. [5] Institute of Psychiatry and Neuroscience of Paris (IPNP), INSERM U1266, Paris, France. [6] Institut Curie, PSL Research University, Centre de Recherche, Laboratoire de Spectrométrie de Masse Protéomique, Paris, France. [7] Sorbonne Université, INSERM, CNRS, Centre d'Immunologie et des Maladies Infectieuses, CIMI-Paris, Paris, France. ✉email: clotilde.thery@curie.fr

All cells release membrane-enclosed vesicles, collectively called Extracellular Vesicles (EVs), in their environment. These EVs contain a selected set of lipids, nucleic acids, and proteins from their cell of origin and thus can transfer a complex array of information to surrounding or distant cells[1,2]. EVs can form by direct outward budding from the plasma membrane (PM) of prokaryotic and eukaryotic cells. In eukaryotic cells, EVs can also form first as intraluminal vesicles of internal multivesicular compartments of the endocytic pathway (MVBs), and are then secreted upon fusion of these compartments with the PM. To clarify the nomenclature, it has been recently recommended to use the term "exosomes" specifically for the MVB-derived EVs, rather than for all small EVs[3]. The PM-derived EVs, on the other hand, are called various names, such as microvesicles, microparticles, or ectosomes: we will use here the latter term which is exclusively used for PM-derived EVs, whereas the others are also used for any type of EVs.

Since they form at different subcellular sites, exosomes and ectosomes will likely contain different sets of specific cargoes, and thus different functions. However, this hypothesis has not been conclusively confirmed so far, given the difficulty to separate exosomes from ectosomes of the same size present in biofluids or cells' conditioned medium, the lack of specific protein markers to distinguish exosomes from ectosomes, and of molecular machineries and tools with demonstrated full specificity for one or the other[4].

Several tetraspanins, especially CD63[5], CD81[5], and CD9[6] have been used as markers of exosomes for the last two decades, due to their accumulation in small EVs as compared to whole cell lysates, and to the steady-state accumulation of CD63 in MVBs. More recently, however, their presence in other EVs has been observed. By capture of EVs bearing specifically either CD63 or CD9 or CD81, followed by analysis of their protein composition and enrichment in endosomal markers, we proposed that EVs bearing only CD9 or CD81 but not CD63 probably did not form in endosomes (and were thus ectosomes), whereas those bearing CD63 together with one or the two other tetraspanins may correspond to endosome-derived exosomes[7]. This observation was made using EVs released by primary human immune dendritic cells, which complicated a direct validation of the model by carrying out cell biology analyses.

This work aims to determine the actual exosomal or ectosomal nature of EVs containing the different tetraspanins. We use HeLa cells, a cellular model amenable to experimental manipulations necessary to address cell biology questions. Our reasoning is that, to identify the subcellular origin of the EVs released by these cells, we need to follow in a time-controlled manner intracellular trafficking of tetraspanins from their initial synthesis in the endoplasmic reticulum (ER) until their secretion in EVs. We have adapted CD63 and CD9 to the Retention Using Selective Hook (RUSH) system, which has been used to follow and control trafficking of numerous transmembrane and secreted proteins, and thus identify atypical pathways of secretion[8]. This allows us also to synchronize the release of CD9- or CD63-containing EVs, to determine the composition of a more homogenous mixture of newly synthesized EVs. Our results demonstrate that both CD63 and CD9 can be released in small ectosomes formed at the plasma membrane, and that in HeLa cells, exosomes represent a minor subpopulation of small EVs (sEVs) that bear CD63 together with other late endosomal molecules such as LAMP1/2. Their secretion is specifically susceptible to neutralization of the endosomal pH. Secretion of ectosomes, for which we identify specific markers in HeLa cells as BSG and SLC3A2, is conversely insensitive to endosomal pH neutralization. Interestingly, CD81, another tetraspanin used commonly as exosome and/or small EV marker, behaves like CD9 rather than like CD63. The markers and molecular mechanisms specific of exosomes versus ectosomes identified here will pave the way for further studies to decipher their respective functions.

## Results

**CD63 and CD9 are present on two distinct and one common populations of EVs.** In dendritic cells, CD63 and CD9 are secreted abundantly in small EVs, where an EV subpopulation bearing CD63 with CD9 and/or CD81 seemed to correspond to endosome-derived exosomes, but both are also detected in larger EVs[7]. We first analyzed their distribution in EVs released by HeLa cells. CD81, CD9, and CD63 can be found in material recovered from HeLa conditioned medium (CM) after 2 h centrifugation at 200Kx$g$ (200 K pellet) (Fig. 1a). By contrast, hardly any signal was detected in the pellets recovered at 2Kx$g$ (2 K) and 10Kx$g$ (10 K), which, in dendritic cell CM, contained large and medium EVs[7]. The total protein content was very low in the 2 K and 10 K pellets, suggesting that HeLa release mainly small and/or light EVs requiring high $g$-force for efficient pelleting. The 200 K pellets also contained the cytosolic protein syntenin, confirming (as recommended by the MISEV2018 guidelines[9]) the presence of membrane-enclosed vesicles. The endoplasmic reticulum (ER) transmembrane protein calnexin was not detected, showing that the HeLa EVs do not originate from the ER. Finally, acetylcholine esterase (AChE), which we recently showed not to be an exosome nor EV marker, but rather a co-isolated serum-derived non-EV component[10] was detected in the 200 K pellets. Analyzed by Transmission Electron Microscopy (TEM), the 200 K pellet contained a majority of vesicles of around 50 nm in diameter, a less abundant population of around 120 nm, and a minor population of about 200 nm (Fig. 1b, right panel). The smallest (50 nm) particles were not detected by Nanoparticle Tracking Analysis (NTA), whose detection limit for biological particles is around 70 nm. By NTA, the 200 K pellets contained a major population of 120 nm, less abundant EVs of 200–300 nm, and few of 300–350 nm in diameter (Fig. 1b, left panel). These results thus confirm that a majority of EVs released by HeLa are smaller than 250 nm, and we will call them here small EVs (sEVs).

To determine whether CD9 and CD63 are on the same or on different sEVs, we immuno-isolated these EVs with antibodies specific for one or the other tetraspanin (Fig. 1c). The HeLa CM was filter-concentrated, rather than ultracentrifuged, to concentrate EVs while avoiding aggregation which can be induced by ultracentrifugation of some biofluids like plasma[11]. CM was obtained in the absence of fetal bovine serum to avoid clogging of the filter concentrator by excess of serum-derived proteins. Side-by-side analysis of the isolated EVs (pull-down: PD) and the EVs that had not bound the antibody (Flow-Through: FT) shows that anti-CD63 co-precipitates around 50% of CD9, and that conversely anti-CD9 co-precipitates around 50% of CD63 (Fig. 1c). Immunoprecipitation of CD63 and CD9 in a mixed conditioned medium from CD63-KO and CD9-KO cell lines shows much lower co-isolation of the other tetraspanin by either antibody (Supplementary Fig. 1a, b). Therefore, the observed co-precipitation of CD63 and CD9 from WT CM is not due to aggregation of single positive EVs but rather due to simultaneous presence of the two tetraspanins on the same EV. This suggests that HeLa cells secrete at least three EV populations defined by these markers: one population with both CD63 and CD9, one with CD63 only and one with CD9 only. Indeed, analysis of the 200 K pellets by immuno-EM after double-labeling for CD9 and CD63 confirmed the existence of these three populations (Fig. 1d). We next questioned the sub-cellular origin of the CD63 + /CD9 + EV population released by HeLa cells. By immunofluorescence, we observed that CD63 is located mainly in intracellular

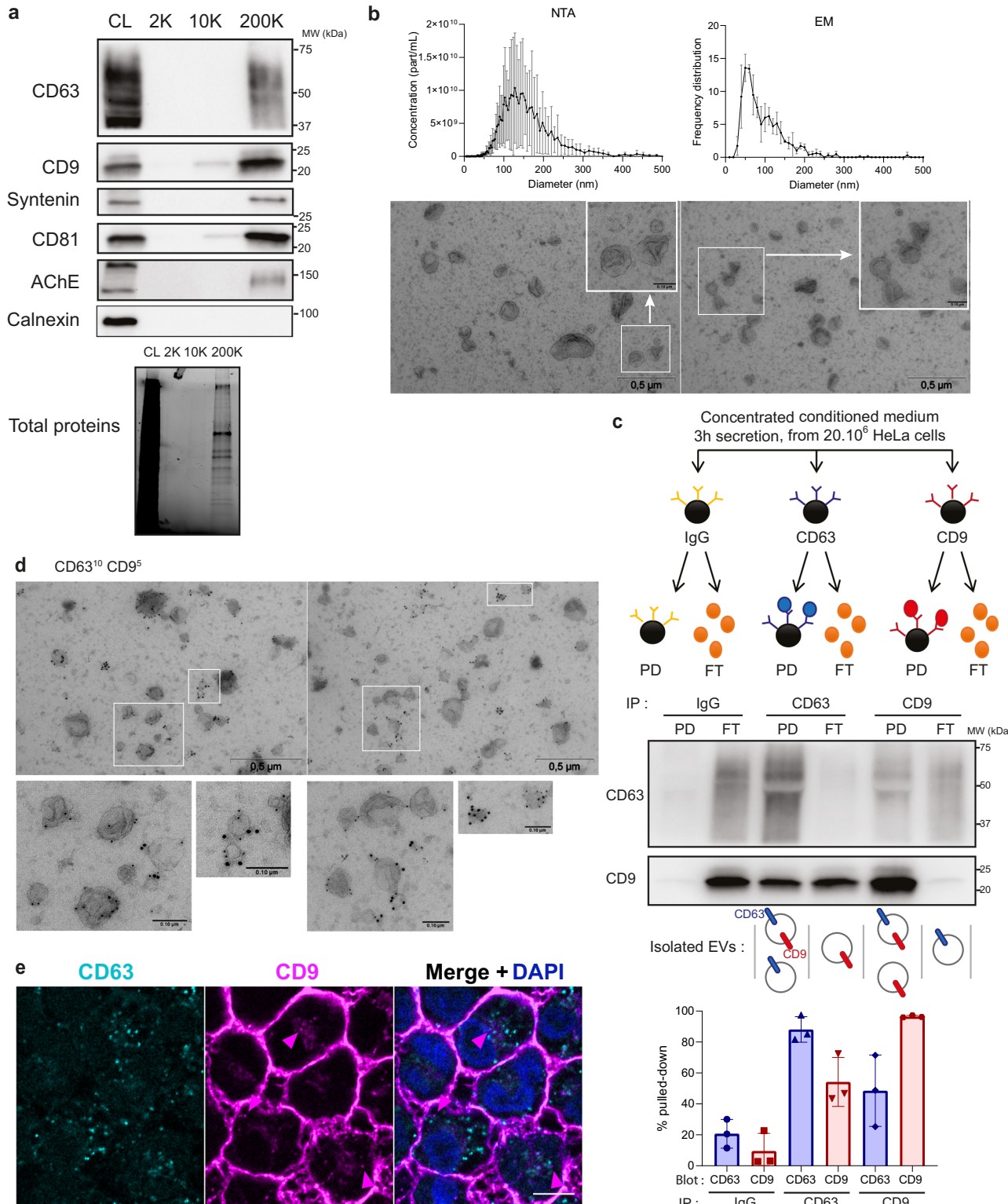

**Fig. 1 CD63 and CD9 are found on two different and one common EV populations. a** Western blot showing transmembrane proteins (CD9, CD63, and CD81), a cytosolic protein (syntenin-1) and two "negative" controls (AChE and calnexin) in cell lysates (CL) and the pellets obtained from HeLa 24 h conditioned medium after differential ultracentrifugation (2 K, 10 K, 200 K). The loaded material comes from $20 \times 10^6$ cells for the centrifugation pellets, and from $0.2 \times 10^6$ cells for the cell lysate. Representative of 3 independent experiments. **b** NTA and EM analysis of 200 K pellets obtained from HeLa 24 h conditioned medium. The quantifications represent the mean of concentration or frequency of EVs of different diameters, error bars show the standard deviation (SD). $N = 3$ independent experiments. Scale bar of the zooms: 0.1 μm. **c** Principle of immunoprecipitation of CD63 and CD9 EVs in HeLa concentrated conditioned medium and representative Western blot of the pull-down (PD) and flow-through (FT) of the immunoprecipitation, with quantification of the relative CD63 and CD9 bands intensity of three independent experiments (mean ± SD is represented). **d** Immuno-EM analysis of the 200 K pellet labeled with anti-CD9 (5 nm gold particles) and anti-CD63 (10 nm gold particles) antibodies. This experiment was performed once. Scale bar of the zooms: 0.1 μm. **e** Confocal imaging of immunofluorescence staining of CD63 and CD9 in HeLa cells. Pink arrows show CD9 localized in intracellular compartments. Representative picture of two independent experiments. Scale bar: 5 μm.

compartments. In contrast, CD9 is mainly found at the plasma membrane but also in rare dim intracellular compartments. No clear co-localization of the two proteins was observed at steady-state (Fig. 1e). Thus, following the dynamics of CD63 and CD9 localization in the cell during their transport is required to understand biogenesis of the double-positive EVs.

**CD63 and CD9 traffic transiently through late endosomes and the PM**. The RUSH system[8] was used to follow CD9 and CD63 synchronously by live imaging from the ER to their residency compartment (Fig. 2a). The principle of this assay is to reversibly retain a protein of interest in a donor compartment, like the ER, and to monitor its release by live imaging. To this end, the protein of interest is fused to a fluorescent moiety and to the streptavidin-binding peptide (SBP), and co-expressed with another protein localized in the donor compartment, fused to streptavidin and used as a hook (here: Streptavidin-KDEL for retention in the ER). Upon synthesis, the SBP-fluorescent protein is retained in the ER by its interaction with streptavidin-KDEL. Biotin is then added to release the protein of interest. Biotin binds to streptavidin and induces the release of the SBP-fused protein that can then follow its normal trafficking route. The fluorescent protein and the SBP were inserted in the small extracellular loop of CD63 and CD9 (Fig. 2a), following a strategy previously used to follow trafficking of multivesicular bodies (MVBs) and their fusion with the PM[12]. Indeed, the CD63-pHLuorin construct used in this previous study behaved similarly to the endogenous CD63 in terms of steady-state intracellular localization and release in EVs, thus showing that this particular site of insertion of the fluorescent protein did not create artefacts for further study of its trafficking.

The trafficking of the two tetraspanins was quantified and compared in HeLa transiently co-transfected with the RUSH plasmids coding for CD63-mCherry and CD9-eGFP (Fig. 2b). Between 15 and 40 min after leaving the ER, both enter and leave the Golgi with similar kinetics. Then they separate (evidenced by a decrease of their Pearson's co-localization coefficient) and CD63 eventually accumulates in intracellular compartments while CD9 reaches the cell periphery. However, between 30 min and 1 h after biotin addition, after leaving the Golgi, a fraction of both proteins was found in some common internal compartments (Fig. 2b, white arrowheads). Adding NH₄Cl, which neutralizes the pH of acidic compartments, to cells transfected with the RUSH plasmids for either CD63-eGFP or CD9-eGFP led to an increase of fluorescence, stronger for CD63 than for CD9 (Supplementary Fig. 2a). This is likely due to unquenching of eGFP which is sensitive to acidic conditions, and this demonstrates that CD63 is going mainly into acidic compartments, but that a fraction of CD9 might also be addressed in such compartments. Immunostaining and EM of eGFP and mCherry in RUSH-CD9 and -CD63 co-transfected cells (Fig. 2c) shows that, at all time points, a majority of MVBs and lysosomes contained only CD63 and not CD9. However, at 1 h and 2 h after biotin addition, some MVBs containing both CD63 and CD9 can be observed (Fig. 2c). CD9 is observed with a Relative Labeling Index (RLI) in MVBs >1 showing a specific location in these compartments. Analysis of HeLa by immunofluorescence after staining of RAB7 confirmed that, 1 h after biotin addition, CD9-eGFP colocalized with CD63-mCherry in late RAB7-positive endosomes to a significantly higher extent than CD9 present in CD63-negative compartments (Fig. 2d). Conversely, CD63 partially and transiently localized at the plasma membrane 1 h and 2 h after biotin addition (CD63 RLI at PM > 1, Fig. 2c). At steady state, the RLI for CD63 on the PM and CD9 in MVBs were both <1 (Fig. 2c), consistently with our IF results on endogenous CD63

and CD9 (Fig. 1e). Some co-localization of the CD63-RUSH construct with MyrPalm-mCherry, which labels the plasma membrane, was also observed between 20 and 60 min post biotin addition (Supplementary Fig. 2b). Two transient places of co-localization of CD63 and CD9 were thus detected: in MVB/lysosomes and at the PM.

**The mutant CD63-YA does not traffic to acidic internal compartments**. Since CD63 can be present at the PM, we then asked if it could be secreted in EVs from this location. To answer this question, we first generated a form mutated in its lysosome targeting motif: CD63-YA (GYEVM - > GAEVM) (Fig. 3a). This mutant has previously been described to interact neither with AP2 nor with AP3, which leads to its plasma membrane localization[13]. Imaging HeLa transfected with the CD63-WT-eGFP or CD63-YA-eGFP RUSH plasmids confirmed that at steady state, contrary to CD63-WT, CD63-YA was located mainly at the cell periphery and in a few small intracellular compartments (Fig. 3b). Immunolabeling with antibodies against EEA1, RAB5, RAB7, and LAMP1, which label endosomes of the different stages of the endocytosis pathway (Supplementary Fig. 3a), showed a similar trend of colocalization of CD63-YA and CD63-WT with all markers, but to a lower overall level for CD63-YA, due to its detection at regions of the PM. In addition, CD63-YA was clearly less co-localized than CD63-WT with LAMP1 (−62%), and with RAB5 (−72%), whereas colocalization with EEA1 and RAB7 was less decreased (−35% and −38%, respectively). This observation suggests that, when inside the cell, and among the markers we tested, CD63-YA is more localized in EEA1- and in RAB7-positive early and late endosomes than other compartments (Supplementary Fig. 3a). CD63-YA arrival and exit from the Golgi showed similar kinetics as those of CD63-WT and CD9 (Supplementary Fig. 3b). CD63-YA then appeared in small intracellular compartments, rather than large ones like CD63 (Supplementary Fig. 3b). When NH₄Cl was added 1 h after biotin addition, only a slight increase of total fluorescence intensity was observed (Supplementary Fig. 3c), similar to the increase observed for CD9-eGFP. This confirms that CD63-YA does not go to acidic compartments as much as CD63-WT. Finally, co-transfecting HeLa with CD63-YA and CD63-WT or CD9 RUSH plasmids showed a similar trafficking for CD63-YA and CD9, with mainly peripheral localization 2 h after biotin addition (Fig. 3c).

**CD9 and CD63-YA have similar kinetics of PM localization and internalization, which differ from those of CD63**. To confirm that the different tetraspanins use the PM as transport intermediate, and quantify more precisely the extent and kinetics of such transport, we quantified PM exposure carrying out anti-GFP staining of non-permeabilized HeLa transfected with CD63-WT-eGFP, CD63-YA-eGFP, or CD9-eGFP RUSH plasmids. The relative expression of GFP at the surface (anti-GFP-AF647 signal normalized to total GFP expression in fixed cells) was then quantified by flow cytometry at different times after biotin addition and at steady state corresponding to continuous presence of biotin from the time of transfection (Fig. 4a, Supplementary Fig. 4). CD9 and CD63-YA reached a similar surface (AF647)/total (GFP) fluorescence intensity ratio of 1 around 1 h after biotin addition, which increased further to around 2 at steady state. For CD63, by contrast, the maximum surface/total ratio increased but remained below 1 until 2 h post biotin addition, and decreased between 2 h and the steady-state (Fig. 4a). This kinetic quantitative analysis thus confirms the microscopy observations: a portion of CD63 is transiently localized at the plasma membrane, whereas CD63-YA and CD9 behave similarly

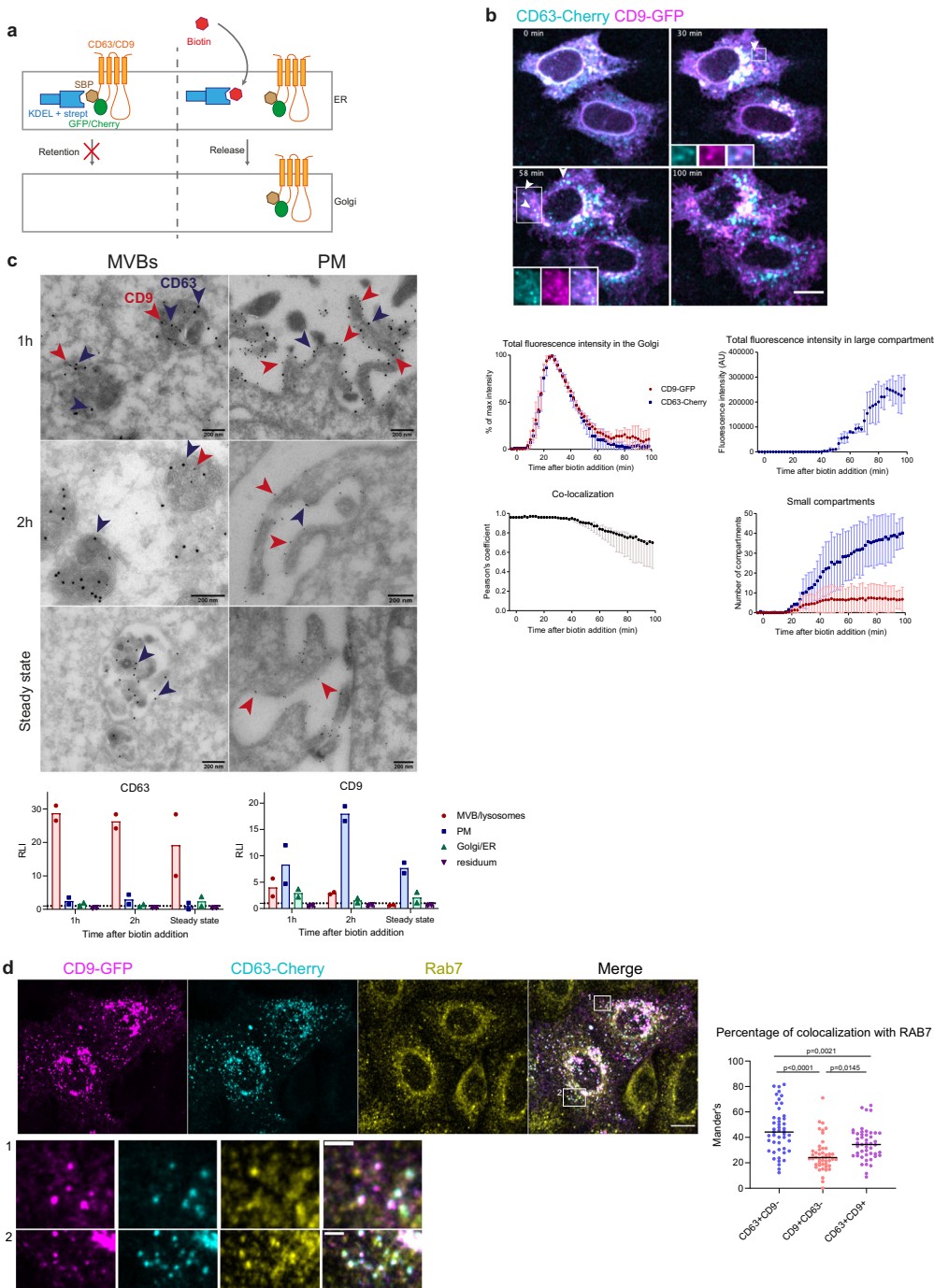

**Fig. 2 CD63 and CD9 transiently co-localize in multivesicular bodies and at the plasma membrane. a** Principle of the RUSH system used to follow CD63 and CD9 intracellular trafficking. SBP streptavidin binding peptide, strept streptavidin, ER endoplasmic reticulum. **b** Micrographs and quantifications of live imaging of HeLa cells co-transfected with the *CD63-mCherry* and *CD9-eGFP* RUSH plasmids. Biotin at 40 µM was added at $T = 0$. White arrows show peripheral compartments where CD63 and CD9 co-localize. *Z*-projection of 11 planes. Scale bar: 5 µm. Quantification upon time of three independent experiments showing the mean ± SD eGFP and mCherry fluorescence intensity in the Golgi and in large compartments, the mean ± SD number of eGFP- or mCherry-positive small compartments and the median and range of the Pearson's co-localization coefficient between eGFP and mCherry where automatically quantified. $N = 3$ independent experiments. 5 fields per experiments where imaged, for a total of at least 10 individual cells to analyze per experiment. **c** Representative electron microscopy images of HeLa cells co-transfected with RUSH constructs of *CD63-mCherry* and *CD9-eGFP*, 1 h or 2 h after incubation with biotin, or at steady-state, labeled with anti-eGFP gold 10 nm (red arrows) and anti-mCherry gold 15 nm (blue arrows). Relative labeling index (RLI) in each compartment quantified from 7 different fields per replicate is represented as mean ($n = 2$ independent biological replicates). **d** Confocal microscopy pictures of HeLa cells (z-projection) co-transfected with *CD63-mCherry-* and *CD9-eGFP*-RUSH plasmids, and stained with anti-Rab7 after 1 h of incubation with biotin. Scale bar: 10 µm. Mander's coefficients representing the % of CD9 + /CD63-, CD9-/CD63 + , and CD9 + /CD63 + intracellular compartments also positive for the Rab7 signal in each cell are shown. Results from two independent experiments are shown, each dot represents one cell (23 cells from replicate 1, and 24 cells from replicate 2) and the median is represented. Ordinary one-way ANOVA with a Tukey's multiple comparisons test was performed to compare the different categories of intracellular compartments shown on the graph.

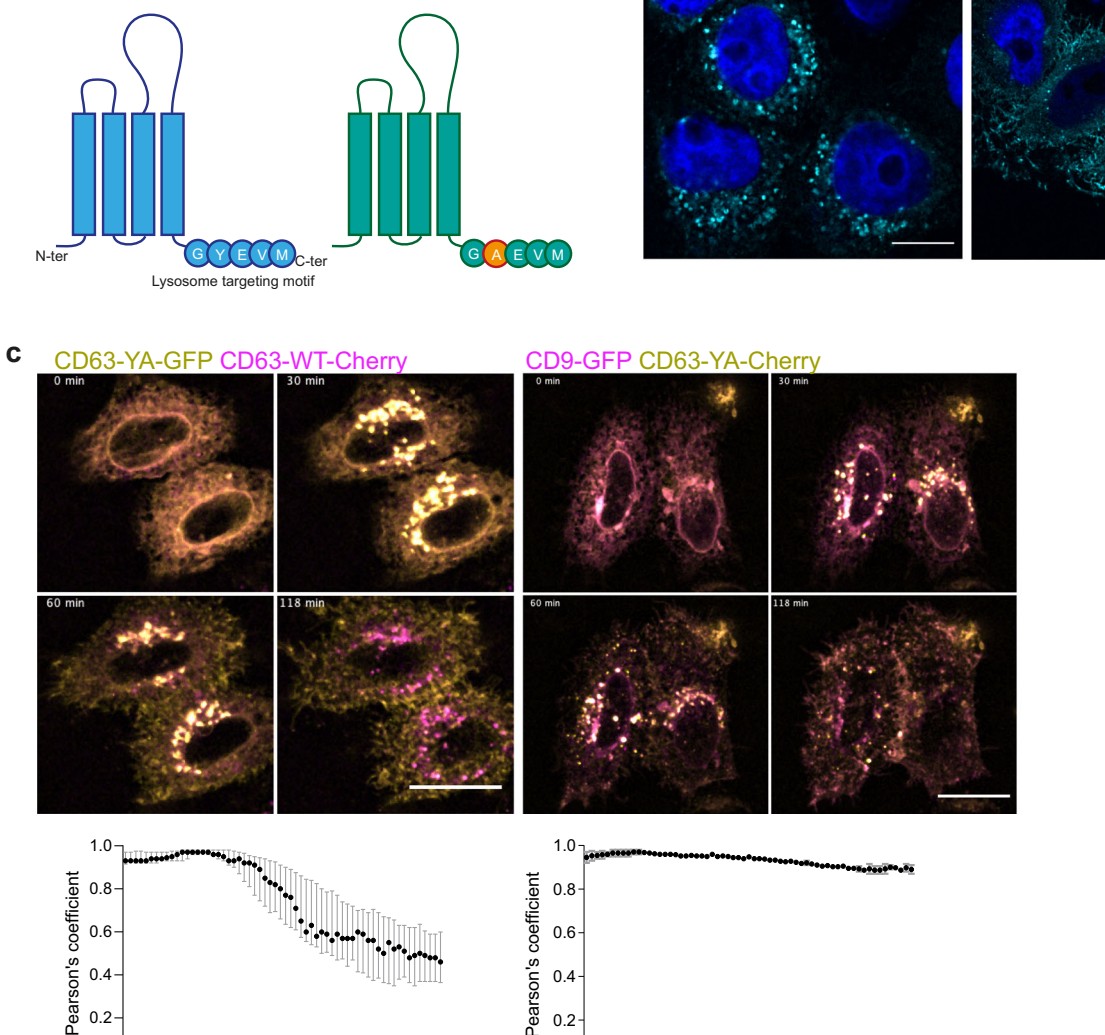

**Fig. 3 The mutant CD63-YA displays different trafficking from CD63-WT and similar trafficking as CD9. a** Scheme of the structure and C-terminal sequences of CD63-WT and the mutant CD63-YA. **b** Immunofluorescence of HeLa cells transfected with the RUSH *CD63-eGFP* or *CD63-YA-eGFP* plasmids at steady state. Scale bar 10 μm. This experiment was performed once. **c** Micrographs of HeLa cells co-transfected with *CD63-YA-eGFP* and *CD63-WT-mCherry* or *CD9-eGFP* and *CD63-YA-mCherry*, and median ± range of the Pearson's co-localization coefficient over time between eGFP and mCherry. Biotin was added at *T* = 0. *Z*-projection of 11 planes. Scale bar 5 μm. CD63/CD63-YA: 3 independent experiments *n* = 51 cells, CD9/CD63-YA: 2 independent experiments *n* = 33 cells. 5 fields per experiment where imaged, for a total of at least 10 individual cells to analyze per experiment.

and accumulate at the plasma membrane. These different behaviors could be due to different rates of internalization from the plasma membrane. To quantify this internalization, we performed an antibody uptake assay: 2 h after biotin addition, cells were incubated with anti-GFP-AF647 antibodies to label RUSH-tetraspanins at the surface, and incubated at 37 °C for 1 h. Cells were then treated by trypsin to remove antibodies still present at the cell surface (= stripping, Fig. 4b) before quantitative analysis by flow cytometry. This treatment efficiently removed antibodies remaining at the cell surface, since the signal was reduced by 77% ± 2.8 for CD63-eGFP, 87% ± 0.4 for CD63-YA-eGFP and 94.9% ± 0.8 for CD9 in cells incubated at 4 °C. While around 80% of CD63 is internalized (i.e., the AF647 signal is reduced by 20% in the stripped condition), only 30% of CD9 and 40% of CD63-YA are internalized (Fig. 4b). Collectively, the live tracking and cell

surface arrival experiments show that CD63-YA behaves mostly like CD9: it accumulates at the cell surface, but a minor portion also undergoes re-internalization, whereas the WT CD63 is quickly and massively re-internalized, as previously shown[13].

**EVs released by HeLa contain more CD9 and CD63-YA than CD63.** We then asked whether altering subcellular localization of the same molecule, CD63, affects its release in EVs. After 24 h of treatment with biotin, the protein synthesized from the transfected RUSH *CD63-YA-mCherry* was efficiently released in small EVs (200 K pellet, Fig. 5a). Interestingly, more mCherry signal was detected in EVs from *CD63-YA* than *CD63-WT*-transfected cells. This difference was even more striking when EVs released during 24 h with biotin were recovered by anti-GFP

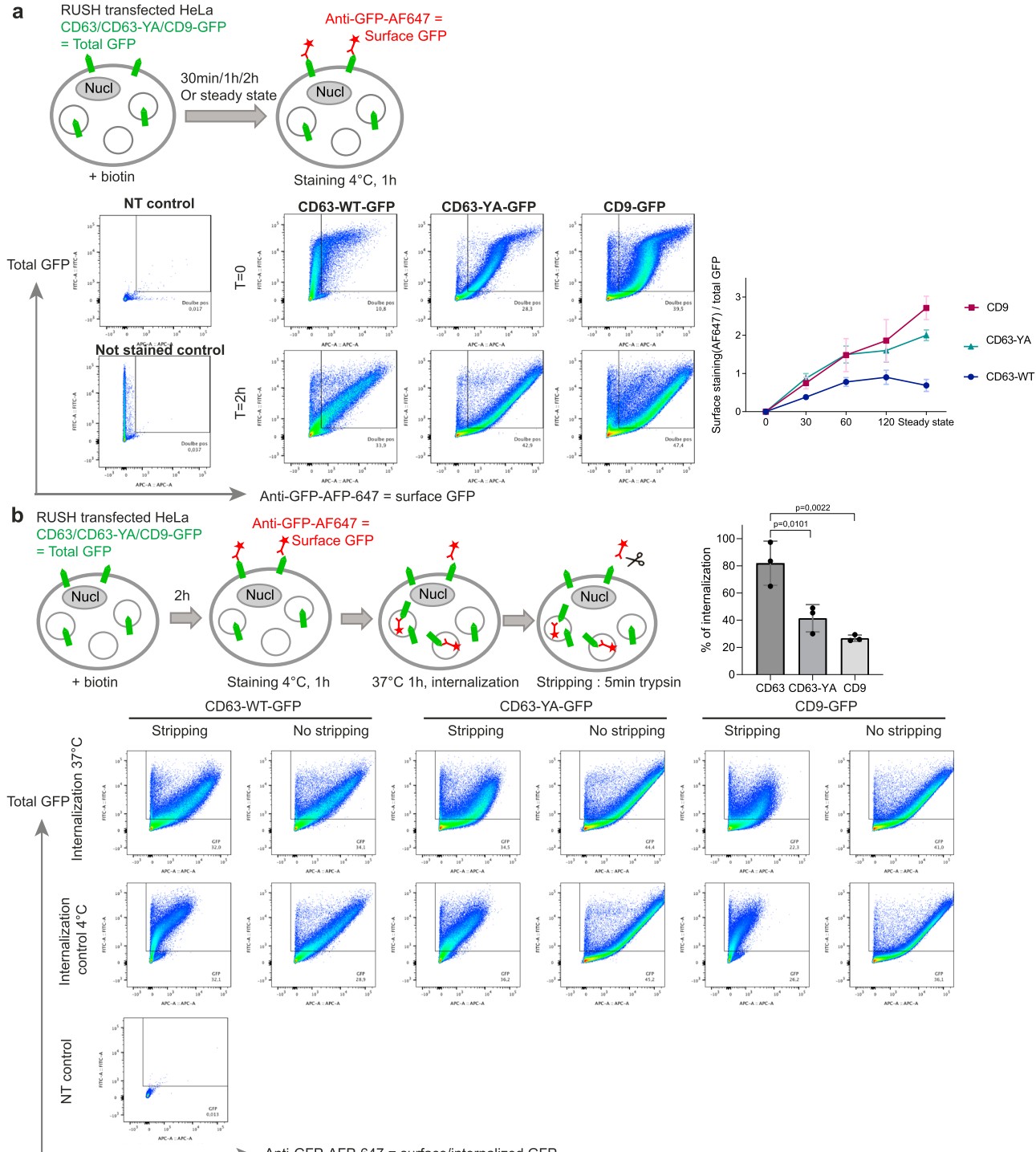

**Fig. 4 Dynamics of localization at the plasma membrane and of internalization of CD63 are different from those of CD63-YA and CD9. a** Principle and flow cytometry analysis of anti-GFP surface staining of HeLa cells transfected with the RUSH constructs *CD63-WT-eGFP*, *CD63-YA-eGFP*, or *CD9-eGFP* after different incubation times with biotin followed by fixation. The ratio of the surface staining (AF647) over the total GFP signal mean fluorescence intensities is represented at different time points, time 0 subtracted, mean ± SD for 3 independent experiments. **b** Principle and flow cytometry analysis of anti-GFP uptake after surface staining of HeLa cells transfected with the RUSH constructs *CD63-WT-eGFP*, *CD63-YA-eGFP*, or *CD9-eGFP* after 2 h of incubation with biotin. The mean percentage ± SD of internalized anti-GFP-AFP647 is represented for 3 independent experiments. Ordinary one-way ANOVA, Tukey's multiple comparisons test. Gating strategy is illustrated in Supplementary Fig. 4.

immuno-precipitation from CM of HeLa cells transfected with *CD9-*, *CD63-WT-*, or *CD63-YA-eGFP* RUSH constructs (Fig. 5b). While similar percent of cells expressed GFP upon transfection with the 3 RUSH-constructs (Supplementary Fig. 5a), and similar numbers of EVs of similar sizes were released by the RUSH

constructs-transfected cells (Supplementary Fig. 5b, c), between 2 and 4 times more GFP signal was detected per particle for both CD63-YA and CD9 than for CD63-WT (Fig. 5b).

BafilomycinA1 (BafA1) is a vacuolar ATPase inhibitor which inhibits the acidification of late endosomes and has been

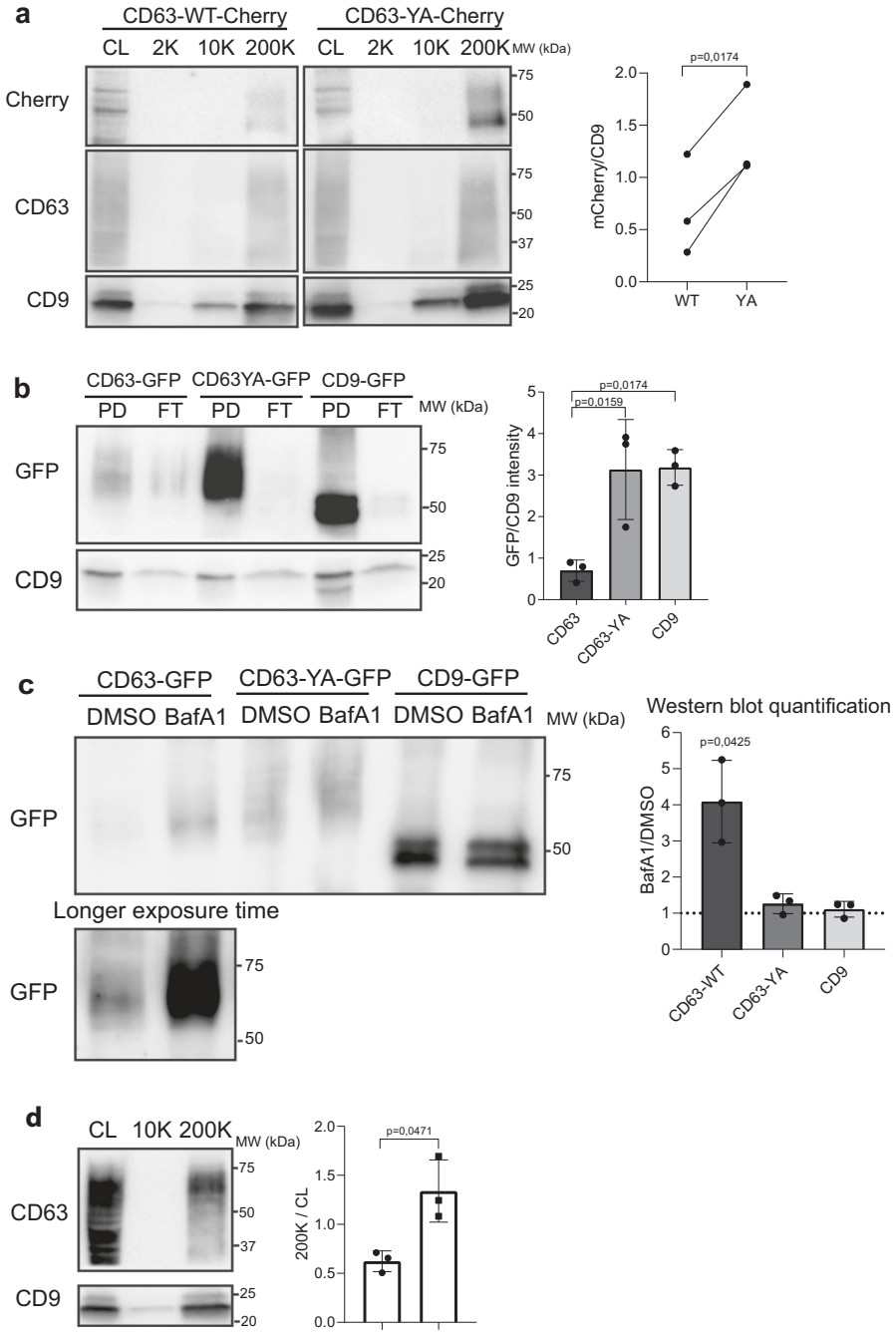

**Fig. 5 CD63-WT is less efficiently released in EVs than CD9 and CD63-YA. a** Western blot of the cell lysate (CL) and of the different EV pellets obtained by differential ultracentrifugation of CCM from HeLa cells transfected with the RUSH plasmids *CD63-WT-eGFP* or *CD63-YA-mCherry* (24 h release with biotin). EVs from $20 \times 10^6$ cells and CL from $0.2 \times 10^6$ cells were loaded. The intensity of the band corresponding to the mCherry fusion proteins was quantified and normalized by the intensity of the CD9 band in 3 independent experiments, the mean ± SD is represented. Two-tailed paired *t* test.
**b** Representative Western blot of the pull-down (PD) and flow-through (FT) of the immunoprecipitation of EVs from HeLa cells transfected with the RUSH plasmids *CD63-WT-eGFP*, *CD63-YA-eGFP*, or *CD9-eGFP*, recovered 24 h after biotin addition. $60 \times 10^8$ total particles quantified by NTA were used for each IP. Percent of GFP + cells quantified by flow cytometry were similar in the three conditions (Supplementary Fig. 5a). The GFP bands intensity in the PD normalized to endogenous CD9 in the corresponding PD are represented as mean ± SD for 3 independent experiments. Ordinary one-way ANOVA, Tukey's multiple comparisons test. **c** Representative Western blot of EVs (200 K pellets) from HeLa cells transfected with the *CD63-WT*, *CD63-YA*, or *CD9-eGFP* RUSH plasmids treated with DMSO of BafA1 100 nM during 16 h. The same number of EVs between the DMSO and the BafA1 conditions were loaded on the gel (around $100 \times 10^8$ particles). The fold change between DMSO and BafA1 treatment for each construct is represented as mean ± SD for 3 independent experiments. Two-tailed one sample *t* test to compare each condition with a theoretical mean of 1. **d** Proportion of cellular endogenous CD9 and CD63 released in EVs, as semi-quantified on Western blots. The signal for CD9 and CD63 in 200 K pellets released by $20 \times 10^6$ HeLa cells was divided by the signal for the same molecule in the total lysate of $0.2 \times 10^6$ cells, run on the same blot. 1 representative Western blot and quantification (mean ± SD) of 3 independent experiments. Two-tailed paired *t* test.

described to stimulate the secretion of CD63 + EVs[14,15] and we confirmed that CD63-eGFP release in EVs was strongly increased by BafA1 (Fig. 5c). In contrast, only minor increases in CD63-YA-eGFP or CD9-eGFP content of EVs were observed upon BafA1 treatment (Fig. 5c), while the total amount of released EVs was increased for all cells (Supplementary Fig. 5d). These results show that CD63 can be efficiently released in small EVs when its location is stabilized at the plasma membrane through mutation of its endocytosis signal, and confirm that different mechanisms are responsible for the release of CD63 in EVs depending on its localization in endosomes or at the PM. They also suggest more abundant release of CD9 and the PM-localized mutant CD63 in EVs than the MVB-enriched WT CD63. Consistent with this hypothesis, in the non-transfected HeLa cells, although both CD9 and CD63 are detected in EV pellets, the ratio of total signal in pellets versus whole cell lysates was always lower for CD63 than for CD9, also suggesting a lower efficiency of release of CD63-bearing EVs (Fig. 5d). Collectively, these results suggest that in HeLa cells, sEVs are more likely to be secreted from the PM than from MVB under steady-state conditions.

**Quantitative mass spectrometry analysis of synchronously secreted CD63 or CD9 EVs reveals signatures of plasma membrane- and endosome-derived EVs.** To better determine the origin of CD63 + and CD9 + EVs and to identify specific markers of EVs released from PM or from endosomes, we performed a quantitative mass spectrometry analysis of GFP + EVs recovered by immunoprecipitation from HeLa transfected with the *CD63-WT-eGFP* or *CD9-eGFP* RUSH plasmids, incubated during 3 h or 24 h with biotin. The 3 h time point was chosen to analyze freshly released EVs, while allowing sufficient recovery of EV proteins for proteomic analysis. The 24 h time point matched a classical timing used to prepare conditioned medium for EV isolation. At 3 h after biotin addition, CD63-mCherry and CD9-eGFP displayed similar low colocalization level, quantified on fixed cells, as at 2 h, and slightly higher than at 24 h (Supplementary Fig. 5e).

The RUSH system had several unique advantages in this approach. First, the analysis of EVs after short-term trafficking of the RUSH constructs enabled to study a majority of freshly secreted EVs avoiding too many cycles of re-uptake and recycling. Second, use of the same anti-eGFP antibody to isolate both RUSH construct-bearing EVs ensured identical efficacies of immuno-isolation, which is necessary for reliable quantitative comparison (and difficult to achieve when using different antibodies, e.g., specific for either CD9 or CD63). Third, specificity of release of the identified proteins with CD63 or CD9 was verified by simultaneously analyzing immunoprecipitated negative control samples containing EVs released by non-transfected cells.

A total of 333 and 397 proteins specifically isolated by the anti-GFP IP were quantitatively compared between CD63-eGFP and/ or CD9-eGFP EVs at, respectively, 3 h and 24 h (Supplementary dataset 1, Supplementary dataset 2). They were categorized as enriched (over two-fold) or unique in CD63- or CD9-bearing EVs, versus common to CD63- and CD9-bearing EVs (Fig. 6a, see Materials and methods for detailed criteria).

To try to determine the subcellular origin of these EVs, we compared our protein lists with a reference database of >5000 proteins specifically assigned to HeLa subcellular compartments[16] using the FunRich software[17,18]. For each category of proteins (enriched in CD63-EVs, enriched in CD9-EVs or common to CD63- and CD9-EVs), we calculated the percent assigned to the different intracellular organelles and whether this percent was significantly different from that of the reference database of intracellular proteome. At 3 h (Fig. 6b, left), the CD63/CD9

common category was significantly enriched for PM proteins (14.8% versus 9.7% in the total proteome), whereas this enrichment was not observed for either CD63- or CD9-EV-specific proteins (7 and 5%, respectively). A significant enrichment was observed for lysosomal proteins in both CD63-EVs and in the CD63/CD9 common category (7 and 5.7% versus 1.7% in the total proteome). Interestingly, transmembrane lysosomal proteins (LAMP1, LAMP2, PLD3) were only found in CD63-EVs, whereas only soluble lysosomal enzymes (CTSV/D/C and TPP1) were enriched in the CD63/CD9 common category (Table 1). This suggests that a specific sorting of transmembrane lysosomal cargoes into EVs by CD63 may occur, without involvement of CD9, whereas the CD63- and/or CD9-EVs trafficking into lysosomes would capture on their surface some luminal proteins. At 24 h (Fig. 6b, right), we observed in CD9-EVs a much stronger and significant enrichment of PM components (28.3%), than in the common CD9/CD63 category or the CD63-EVs (respectively, 10% and 2%). Lysosomal protein enrichment decreased slightly but was still significantly higher than that in the total proteome, in both the CD63-EVs and the CD63/CD9-common category, and LAMP1 was enriched in the latter (Fig. 6c, Table 1). Thus in 24 h conditioned media, a time routinely used to recover EVs, CD9-bearing EVs come from the PM, whereas CD63/CD9-EVs come from both PM and lysosomes.

We then searched for transmembrane (TM) proteins, using both the GO term analysis and manual annotation, which could associate specifically to one or the other EV type and could be used to label or isolate them by antibodies recognizing their extracellular domains. Only 6–14 TM proteins were present in the CD9-EVs (respectively 3–24 h), 8–8 in the common CD63/CD9-category, and 18–16 in the CD63-EVs (red text in Supplementary dataset 2). Among these, proteins assigned to lysosomes, endosomes (considering all stages of the endocytic pathway: early, recycling, and late endosomes including MVBs) or the PM are shown in Fig. 6c, Table 1. Only PM proteins were enriched in CD9-EVs, whereas both endosome/lysosome and PM transmembrane proteins were enriched in CD63-EVs.

**BafA1 and GW4869 drugs affect differently secretion of CD63 and LAMP1, versus CD9, BSG, and SLC3A2.** We selected for further analysis: *LAMP1* (or CD107a) for its known lysosomal steady-state localization and its enrichment in CD63-EVs and/ or in the common CD63/CD9 category, basigin (*BSG*, a plasma membrane protein also called EMPRINN or CD147) for its different association to CD63 at 3 h and CD9 at 24 h, and the PM 4F2 cell surface antigen heavy chain (*SLC3A2* or CD98), for its common presence in CD63- and CD9-EVs at 3 h and its enrichment in CD9-EVs at 24 h (Fig. 6a).

We first asked how these markers behaved, in terms of enrichment in the 200 K pellets versus cell lysates of non-transfected HeLa cells (Fig. 7a). As observed before (Fig. 4d), CD63 was less enriched than CD9 in the 200 K pellets recovered from 24 h CM. LAMP1 was even less enriched than CD63, whereas both BSG and SLC3A2 were as enriched as CD9, in accordance with their preferential association with CD9-EVs in the proteomic analysis.

We then studied the effects of two drugs used in the literature to disturb exosome secretion (BafA1[14,15] and the inhibitor of neutral sphingomyelinase GW4869[19]) on the secretion of our selected proteins, other sEV markers (syntenin-1/*SDCBP* and CD81), and negative controls (AChE and Calnexin). A 16 h treatment of cells during the medium conditioning period was chosen, as it did not induce loss of cell viability (Fig. 7b). GW4869 treatment decreased in a variable manner the total number of released particles in the 200 K pellet, while it increased the

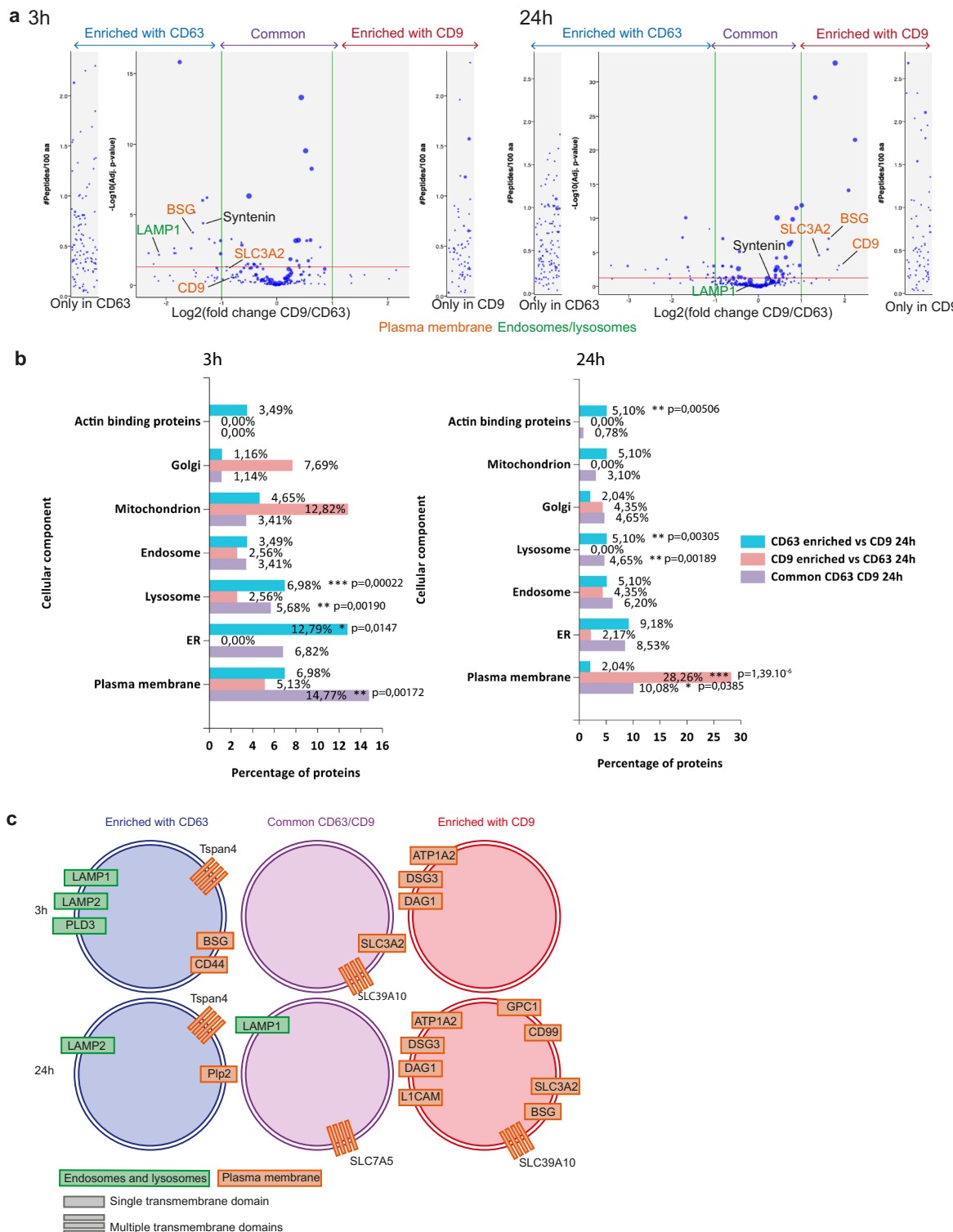

**Fig. 6 Quantitative proteomic analysis of CD63-eGFP-RUSH and CD9-eGFP-RUSH EVs.** EVs were isolated by anti-GFP immuno-isolation from either non-transfected HeLa cells, or HeLa transfected with *CD63-eGFP*-RUSH or *CD9-eGFP*-RUSH, either 3 h or 24 h after biotin addition, and their composition was analyzed by mass-spectrometry. **a** Volcano plots representing quantified proteins with at least 2 peptides in 2 replicates in at least one condition. Shown are the fold changes of peptide abundance between CD63- and CD9-eGFP expressing EV samples and the *p*-value of this quantification, for EVs recovered 3 h (left) or 24 h (right) after biotin addition. Position of the membrane-associated proteins selected for further analysis is indicated. **b** Results of the FunRich gene enrichment analysis among the proteins either enriched in the CD63- (blue), or CD9-eGFP (red) samples, or common between the CD63 and CD9-eGFP samples (purple) at 3 h or 24 h after biotin addition. For each subcellular compartment protein list, % of proteins of this category in the list of CD63-, CD9-, or common CD63/CD9 proteins is indicated, and the *p*-value of this percentage being different to its counterpart in the whole HeLa cell database is calculated. *P*-value (hypergeometric uncorrected). **c** Schematic representation of the transmembrane proteins identified in the CD63-, CD9-, or CD63/CD9-eGFP EVs at 3 h and 24 h.

**Table 1 Names of proteins identified in EVs and used for the endosome, lysosome, and PM (plasma membrane) categories are listed.**

| 3 h | | | 24 h | | |
|---|---|---|---|---|---|
| CD63 enriched | CD9 enriched | Common | CD63 enriched | CD9 enriched | Common |
| **Lysosomes** | | | | | |
| LAMP1* | GBA | CTSV | DNASE2 | | HEXB |
| RNASET2 | | CTSD | RNASET2 | | TPP1 |
| PLD3* | | TPP1 | CD63* | | CTSC |
| CTSZ | | CTSC | SCPEP1 | | CTSD |
| DNASE2 | | HEXB | LAMP2* | | LAMP1* |
| LAMP2* | | | | | CTSV |
| **Endosomes** | | | | | |
| CHMP4B | IST1 | BLMH | CHMP1B | IST1 | CPNE8 |
| ATP6V1D | | TOLLIP | VPS26A | EHD4 | BLMH |
| ATP6V1G1 | | CPNE8 | ATP6V1D | | TOLLIP |
| | | | ATP6V1G1 | | CHMP4B |
| | | | THBS1 | | PRSS23 |
| | | | | | GM2A |
| | | | | | EHD1 |
| | | | | | COL5A1 |
| **Plasma membrane** | | | | | |
| IGFBP7 | DDAH1 | JUP | TSPAN4* | BASP1 | JUP |
| BSG* | DAG1* | S100A14 | PLP2* | CD9* | SLC7A5* |
| TSPAN4* | | ALDOA | | GNG12 | IGFBP7 |
| CD44* | | BASP1 | | B2M | ANXA2 |
| LMNB1 | | ANXA2 | | BSG* | BANF1 |
| TLN1 | | VCP | | SLC3A2* | EZR |
| | | LMNA | | SLC39A10* | S100A10 |
| | | BANF1 | | DDAH1 | ALDOA |
| | | B2M | | GNAI3 | MSN |
| | | CD9* | | CD99* | S100A14 |
| | | CALM1 | | DAG1* | LMNA |
| | | SLC39A10* | | GPC1* | VCP |
| | | SLC3A2* | | L1CAM* | CALM1 |

*Membrane associated proteins.

exosomes, which also contain syntenin-1. In contrast, CD9, BSG-, and SLC3A2-containing EVs are released primarily from the PM. Of note, CD81 like CD9 was only slightly increased by BafA1-mediated endosome alkalization, suggesting that, like CD9, CD81 buds mainly in small ectosomes from the PM. In fact, we observed, by the RUSH approach, a similar trafficking pattern for CD9 and CD81, with final colocalization at the PM (Fig. 8). The BafA1 treatment of EV-secreting cells thus represents a convincing way to evaluate the association of given EV markers to exosomes (increased secretion) versus ectosomes (no or minor increase of secretion).

## Discussion

In this work, we provide evidence that sEVs bearing tetraspanins, especially CD9 and CD81 with little CD63, bud mainly from the plasma membrane, whereas others bearing CD63 with little CD9 but containing some late endosome proteins form in internal compartments and qualify as exosomes. We also identified a small set of additional transmembrane proteins that can be used to distinguish the small ectosomes from bona fide exosomes. To obtain these evidences, we followed in live cells the intracellular trafficking of the tetraspanins and identified their colocalization and segregation over time (Figs. 2–4), we quantified the release in EVs of an endocytosis-defective mutant form of CD63 which traffics like CD9 (Figs. 3–5), we performed quantitative proteomic analysis to identify proteins that are specifically released in EVs with CD9 or CD63 (Fig. 6, Table 1), and we quantified the effect on EV secretion of a drug known to increase the pH of late endosomes, BafA1, to evidence distinct behavior of endosomal versus PM EV markers (Fig. 7).

Incidentally, our study revealed that a drug commonly used as an exosome inhibitor, GW4869, which inhibits neutral sphingo-myelinase and thus prevents ceramide accumulation, should be used with caution. The low or variable effect of GW4869 on release of our EV markers questions its specificity for exosomes. This drug might affect a specific subpopulation of EVs containing syntenin-1 and LAMP1, but not CD63 in HeLa. In addition, the requirement of ceramide for EV secretion largely depends as well on the cell type. For instance, in the breast carcinoma cell line SKBR3, GW4869 increased release of ectosomes while it decreased the amount of small EVs[21]. Accordingly, we observed slightly higher levels of large EVs released by GW4869-treated HeLa cells. Since ceramide accumulation can lead to many additional effects than changing budding at the PM or in endosomes, for instance by promoting apoptosis, interpretation of GW4869 effects of EV release is always difficult.

Comparison of small EVs (containing a mixture of exosomes and small ectosomes), and larger EVs/ectosomes showed differences in their respective protein composition and oncogenic activities[22,23], but these studies did not explore the diversity within small EVs. More recently, further approaches to separate subtypes of small EVs and compare their cargoes have been published[7,24,25]. Importantly, the two most recent studies demonstrated the presence of non-EV components, called extracellular nanoparticles[25], and small lipidic structures called exomeres of unclear vesicular nature[24] within the bulk of small EV preparations. Subtypes of small EVs of different sizes within the 50–150 nm range[24], or slightly different densities in gradients[7,25], were also reported. However, the subcellular origin of these different small EVs was not specifically evaluated, and an exosomal nature could only be speculated, based on enriched presence of molecules known to associate with endosomes in some of the recovered small EVs. Our study has the advantage of combining the use of several markers from endosomes and lysosomes or from the PM with a search for mechanisms involved

particle numbers in the 10 K pellet, as quantified by NTA (Fig. 7c). The size distribution of EVs, analyzed both by NTA and EM (Fig. 7d) was identical in 200 K pellets from control (DMSO-treated) and GW4869-treated cells. Accordingly, GW4869 had a mild or no effect on release of most of the selected EV markers, as analyzed by Western blot (Fig. 7e). Only syntenin-1 and LAMP1 were always decreased by this drug, although with high variability (Fig. 7e). Therefore, GW4869 decreases release in EVs of syntenin-1 and LAMP1, but not the other proteins, including CD63, and it conversely tends to increase release of larger/heavier EVs, as previously observed in other tumor cells[20]. On the other hand, the effect of BafA1 was more clear-cut. HeLa cells exposed for 16 h to BafA1 released almost 4 times more particles in the 200 K (average Log2(Fold change) = 1.8, Fig. 7c). These EVs were smaller, as measured by NTA and EM (Fig. 7c, d). By contrast, the effect was unclear on EVs recovered in the 10 K pellet (Fig. 7c). Molecularly, BafA1 treatment resulted in the release of, respectively, 8 or 32 times more of CD63-syntenin-1 and LAMP1, but less than 2 times more CD81 and CD9. BafA1 either had no significant effect or decreased release of SLC3A2 and BSG, and of AChE and Calnexin, as compared to control cells (measured by Western blot, Fig. 7e). Together with the specific effect of BafA1 on the release of CD63 and not of CD63-YA (Fig. 5c), these results thus confirm that the CD63- and LAMP1-positive subpopulations of EVs represent MVB/lysosome-derived

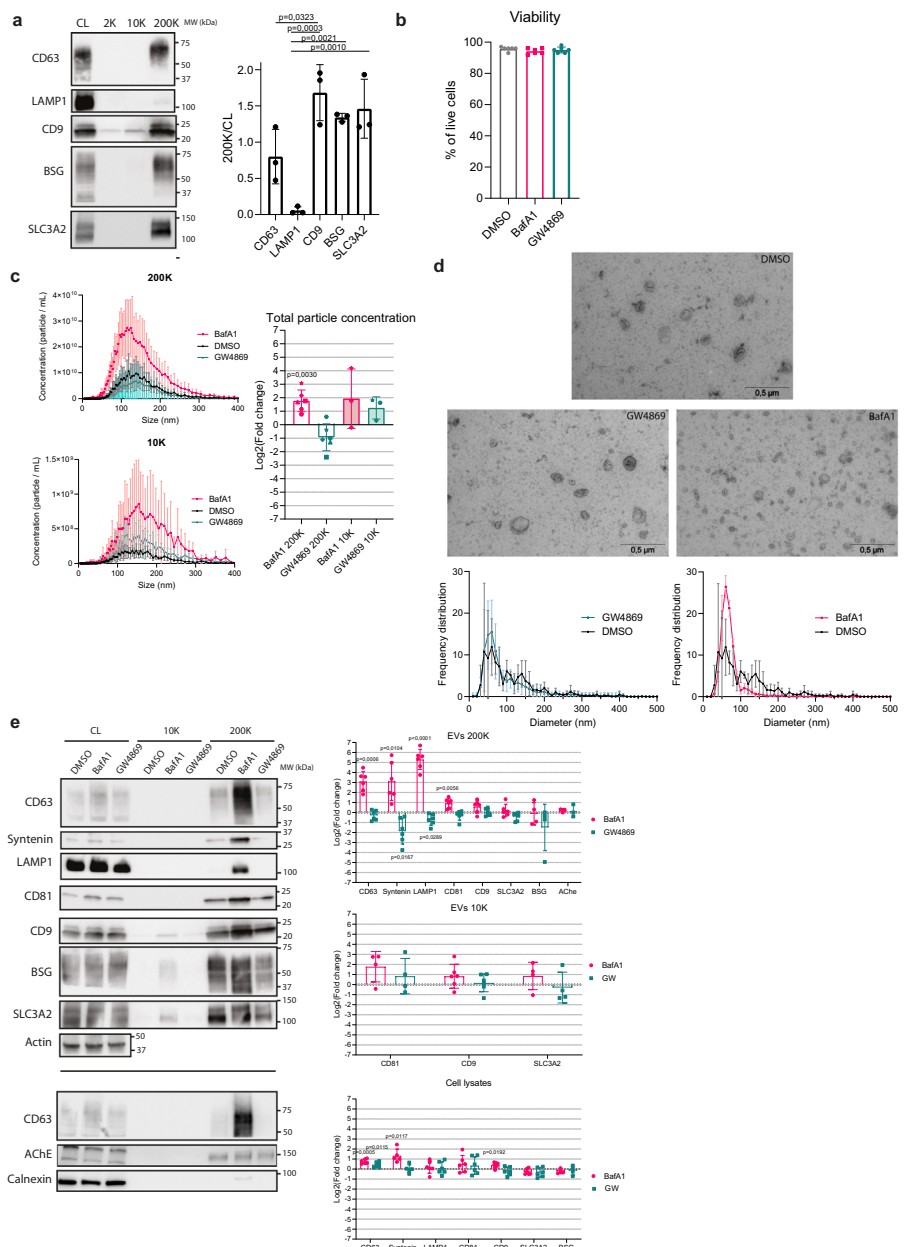

**Fig. 7 Different effect of BafA1 and GW4869 on secretion of CD63, CD9, and the novel EV markers. a** Western blot showing CD9, CD63, and CD81, and the new markers LAMP1, BSG, and SLC3A2 in cell lysates (CL) and the pellets obtained from HeLa conditioned media after differential ultracentrifugation (2 K, 10 K, and 200 K). The loaded material comes from $20 \times 10^6$ cells for the centrifugation pellets, and from $0.2 \times 10^6$ cells for the cell lysate. One representative image. For each marker, mean ± SD of the quantification of the signal in 200 K pellets divided by the signal in the total lysate, run on the same blot, is shown for 3 independent experiments. Ordinary one-way ANOVA, Tukey's multiple comparisons test. **b** Viability of HeLa cells at the end of the 16 h medium conditioning period in the presence of DMSO (control) or BafA1 (100 nM) or GW4869 (10 μM) drugs, measured by trypan blue in 6 independent experiments, mean ± SD is represented. No significant difference observed with an ordinary one-way ANOVA, Tukey's multiple comparisons test. **c** Nanoparticle tracking analysis (NTA) of EVs obtained by differential ultracentrifugation from equal numbers of HeLa cells treated with DMSO (control), BafA1 or GW4869 during 16 h. The particles concentration according to their size and the fold change of the total particle concentration between treated and control conditions are represented as mean ± SD of 5 (200 K) or 3 (10 K) independent experiments. Two-tailed one sample $t$ test to compare each condition with a theoretical mean of 0. **d** TEM analysis (1 representative image) and size measurement (in 3 independent experiments) of EVs in 200 K pellets of cells exposed to DMSO, BafA1 or GW4869. Mean ± SD of the frequency distribution of CD63 and CD9 in EVs of different diameters is represented. **e** Representative Western blot of cell lysates from $0.2 \times 10^6$ HeLa cells and EVs from $20 \times 10^6$ HeLa cells treated with DMSO, BafA1, or GW4869, corresponding to the samples of **b**–**d**. The mean fold change ± SD between DMSO and BafA1 or GW4869 treatment of the bands intensity in the 200 K and 10 K pellets divided by the cell lysate is represented for 6 independent experiments. Two-tailed one sample $t$ test to compare each condition with a theoretical mean of 0.

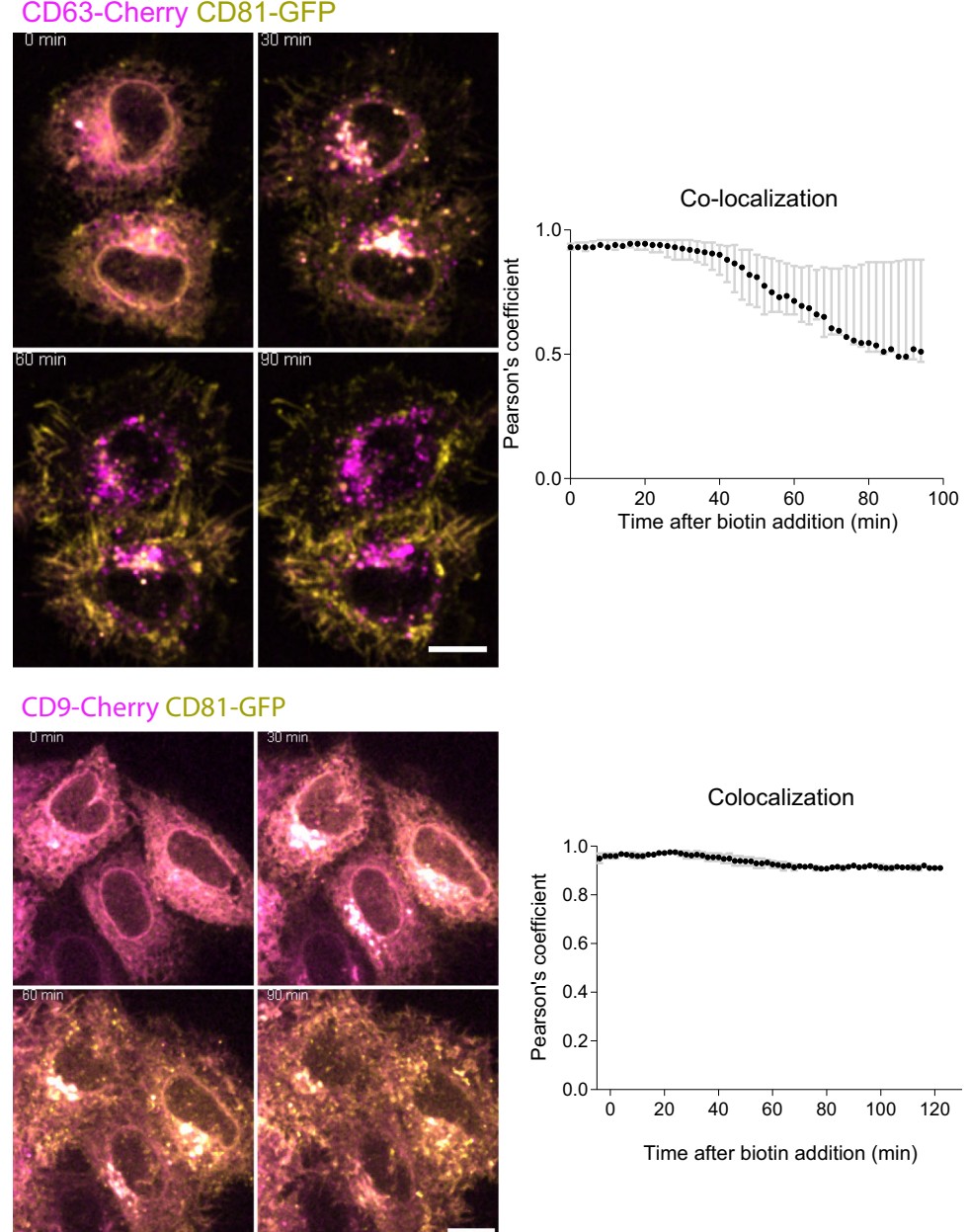

**Fig. 8 Trafficking of RUSH-CD81 compared to RUSH-CD63 and -CD9.** Micrographs of live imaging of HeLa cells co-transfected with either *CD63-mCherry* or *CD9-mCherry* and *CD81-eGFP* RUSH plasmids. Biotin was added at $T = 0$. The median ± range of the Pearson's co-localization coefficient between eGFP and mCherry is represented over time after biotin addition. Scale bar: 10 μm. 2 independent experiments. 5 fields per experiment where imaged, for a total of at least 10 cells to analyze per experiment.

in their secretion by chemical inhibition, to demonstrate the existence of different subpopulations of small EVs, to characterize their protein content and their subcellular origin.

Our results demonstrate that HeLa cells release more small ectosomes than bona fide exosomes. An interesting tool that allowed us to demonstrate this proposal is the mutant CD63 molecule devoid of its lysosome-addressing signal, CD63-YA. CD63-YA accumulates at the PM, like CD9, and is more efficiently released in EVs by HeLa cells than the WT CD63 (Fig. 5). The eGFP-RUSH constructs of CD63 and CD63-YA could now be used as tools to quantify relative secretion of exosomes versus ectosomes in any cell, by transient transfection followed by quantification of the GFP signal recovered from each construct in the conditioned medium. It would be interesting to analyze this way cells secreting at steady-state more CD63 in EVs than HeLa,

and determine whether they indeed display a high ability to secrete bona fide exosomes, or whether instead their CD63-EVs mainly form at the PM.

Our results also led us to propose to use a restricted combination of molecules (CD63, LAMP1, syntenin-1, CD9, CD81, BSG, and SLC3A2) to determine the proportion of ectosomes versus exosomes in a given small EV preparation. Concerning LAMP1, which behaves like CD63 in response to endosomal pH neutralization, its presence on EVs suggests that exosomes can be released from secretory lysosomes as well as from late endosomes. Concerning the PM-EV markers: SLC3A2 is the heavy chain of various heterodimeric amino-acid transporters, and one of its possible partners, Large amino-acid transporter LAT-1 (*SLC7A5* also named 4F2 light chain)[26], was also identified in the proteomic analysis, but was found as a common protein between

CD63 and CD9 EVs at 24 h. BSG and SLC3A2 directly interact together[27], and these two proteins are part of a protein complex interacting with integrins[28]. BSG also interacts with CD44:[29] interestingly, BSG and CD44 were specifically present in CD63-bearing EVs at an early time point of secretion, whereas BSG and SLC3A2 were specific to CD9-EVs at a later time point (Fig. 6c). Thus, BSG is probably released with different types of EVs, depending on its interacting partner. Of note, SLC3A2 is highly expressed in several cancers[30], and BSG has already been described as a cancer cell ectosome marker[21,31], which makes these two proteins good candidates as tumor cell EVs biomarkers.

In addition, a few other markers identified by the proteomic analysis could be in the future also used as either exosome or ectosome-specific markers (Fig. 6). This includes for instance LAMP2, PLD3, PLP2 or the tetraspanin TSPAN4 as exosome markers, and L1CAM, DAG1, CD99, or DSG3 as ectosome markers. Another interesting ectosome marker could be PTGFRN, a known major partner of CD9 and CD81, which has been shown to remain associated with these tetraspanins on EVs secreted by K562 cells:[32] it was identified by 18 peptides with CD9-EVs at 24 h (Supplementary dataset 2), but could not be reliably compared to CD63-EVs because a single common peptide was present in the two samples. BafA1 treatment of cells in combination with quantitative assessment of the resulting EV level of these EV-associated proteins would have to be performed to validate their proposed specificity. Of course, although the action of BafA1 on EVs seems exosome-specific compared to ectosomes in HeLa according to our observations, this should be further demonstrated in other cell types, by similarly assessing an array of EV-associated proteins. In particular, BafA1 can affect also Golgi trafficking:[33] if prominent over the late endosome-specific effect in some cells, this action of BafA1 would be expected to affect more generally secretion and trafficking of most transmembrane proteins, hence the global composition of EVs. Finally, it would be interesting to use other drugs, recently tested in a small screen specifically designed to quantify CD63-EV release[15], to quantify their effect on EV release of the ectosome and exosome markers, and of the mutant CD63-YA reporter.

Our findings and our methodology to discriminate different sEVs subpopulations will be a basis for further studies on the heterogeneity of EVs and their specific functional properties.

## Methods

**Cell culture and transfections**. HeLa cells (obtained from Dr. B. Goud in 1995, and in-house validated by short tandem repeat [STR] analysis in 2018) were cultured in Dulbecco's modified Eagle's medium (DMEM-Glutamax, Gibco), with 10% fetal calf serum (FCS, Gibco), 100 U/mL penicillin, and 100 µg/mL streptomycin (Gibco).

HeLa KO for CD63 or CD9 were generated by CRISPR/Cas9 using sgRNA (sequences provided in Supplementary Table 1) selected using the CRISPR design tool available at the Broad Institute (https://portals.broadinstitute.org/gpp/public/analysis-tools-sgrna-design). The corresponding guide DNA sequences were cloned into the lentiCRISPRv2 plasmid (Addgene #52961) according to the instructions of the Zhang laboratory (https://www.addgene.org/52961)[34]. The plasmids were transfected using Fugene HD according to the manufacturer's instructions, and cells were treated after 36–48 h with 5 µg/ml puromycin for 36–48 h. Cells were stained for CD63 or CD9 and negative cells for the antigen of interest were sorted using a FACS Aria cell sorter (Becton Dickinson). CD63 KO cells were subcloned after two sorting rounds and one clone was kept for experiments. CD9 KO cells were subcloned after two sorting rounds and a pool of 5 clones was used for experiments. CD9 KO cells generated by E. Rubinstein and the CD63 KO clone isolated by R. Palmulli are available upon request to, respectively, C. Théry and G. van Niel (guillaume.van-niel@inserm.fr).

Transient plasmids transfections were performed on 80% confluent cells using the calcium phosphate method: 2.5 ug/mL of plasmids were mixed 1/20 vol/vol with 1 mM Tris pH 8 and 2.5 M CaCl$_2$ and incubated 5 min at room temperature (RT) before being mixed with an equal volume of HEBS (160 mM NaCl, 1.5 mM Na$_2$ HPO$_4$, 50 mM Hepes, pH 7.05). This mix was added at 1/10 vol/vol into culture medium and cells were incubated during 8 h before washing with PBS and replacing with fresh culture media for an additional 16 h before performing experiments.

## Antibodies and reagents

*Primary antibodies for Western blot*. Primary antibodies for Western blot were mouse anti-human CD63 (BD Bioscience, clone H5C6, 1/1000), -human CD9 (Millipore, clone MM2/57, 1/1000), -GFP (Invitrogen, clone GF28R, 1/1000), -human CD147/BSG (Proteintech, clone 1G12B5, 1/10000), -human CD98/SLC3A2 (Proteintech, clone 2B10F5, 1/3000), -human actin (Millipore, clone C4, 1/1000), -human CD81 (Diaclone, clone TS81, 1/1800); rabbit anti-human LAMP1 (GeneTex, clone EPR4204, 1/1000), -mCherry (Biovision, polyclonal, 1/1000), -human calnexin (Abcam, recombinant clone EPR3633(2), 1/1000), goat-anti human acetylcholinesterase (Abcam, polyclonal, ab 31276, 1/500). Monoclonal rabbit anti-human syntenin (1/1000) was a gift from P. Zimmermann (pascale.zimmermann@kuleuven.be).

*Secondary antibodies*. Secondary antibodies HRP-conjugated goat anti-rabbit IgG (H + L) and HRP conjugated goat anti-mouse IgG (H + L) were purchased from Jackson Immuno-Research and used 1/10000.

*Antibodies for immunoprecipitation*. Antibodies for immunoprecipitation were mouse anti-human CD63 (BD Bioscience, clone H5C6, 1/250), mouse anti-human CD9 (Millipore, clone MM2/57, 1/50), normal mouse IgG (Millipore, 1/500); a rabbit anti-GFP polyclonal antibody was produced for the TAb-IP antibody facility and purified by the recombinant protein facility of Institut Curie (for exclusive use by Institut Curie researchers) and used at 1.5 µg/mL.

*Antibodies for immunofluorescence*. Antibodies for immunofluorescence were mouse IgG2b anti-human CD63 (clone TS63b[35] 1/100, available upon request to E. Rubinstein: eric.rubinstein@inserm.fr) and mouse IgG1 anti-human CD9 (clone TS9[36] 1/100) (commercially available at Diaclone or Abcam), mouse IgG1 anti-human EEA1 (BD Transduction Laboratories, clone 14/EEA1, 1/1000), mouse IgG2a anti-human RAB5 (BD Transduction Laboratories, clone 1/Rab5, 1/100), rabbit anti-human RAB7 (Cell Signaling, D95F2, 1/100), mouse IgG1 anti-human LAMP1 (Developmental Studies Hybridoma Bank, H4A3, 1/400), goat anti-mouse IgG2b Alexafluor 647 (Invitrogen, 1/300), goat anti-mouse IgG1 Alexafluor 488 (Invitrogen 1/300), goat anti-mouse IgG Alexafluor 568 (Invitrogen, 1/200), goat anti-rabbit IgG Alexafluor 568 (Invitrogen, 1/200), goat anti-rabbit IgG Alexafluor 647 (Invitrogen, 1/200).

*Antibody for surface staining and uptake experiments*. Mouse anti-GFP-Alexa fluor 647 (BD Biosciences, clone 1A12-6-18, 1/150) was purchased.

*Reagents*. Biotin was purchased from Sigma-Aldrich; a stock solution at 4 mM in DMEM was used at 1/100 in culture medium (40µM final) for all RUSH experiments. BafilomycinA1 ready-made solution 0.16 mM in DMSO was purchased from Sigma-Aldrich and used at 100 nM. GW4869 was purchased from Sigma-Aldrich, dissolved to 5 mM in DMSO, and used at 10 µM. DMSO or drugs were diluted in fresh EV-depleted medium in which cells were cultured for 16 h before collection of conditioned medium.

## Plasmids

**Plasmids**. A RUSH vector encoding for the KDEL-Streptavidin Hook (available upon request to F. Perez: franck.perez@curie.fr) was used for all *CD63*, *CD9*, and *CD81* reporters. A *CD63-SBP-pHluorin* RUSH plasmid was initially generated by inserting the SBP sequence upstream of the pHluorin fragment in the CD63-pHluorin sequence described in ref. [12] (generated by F. J. Verweij and G. Bon-compain and available upon request to G. Van Niel: guillaume.van-niel@inserm.fr). The *CD63-SBP-pHluorin* fragment was synthesized and ligated in a Str-KDEL Neomycin vector after AscI/PacI digestion, thereby replacing the neomycin cassette with *CD63-SBP-pHluorin*. The pHluorin sequence was then replaced by the *eGFP* or the *mCherry* (fragments taken from plasmids described in ref. [8]) sequence (synthesized by Integrated DNA Technologies) to generate RUSH plasmids allowing the expression of human CD63 with the SBP and eGFP or mCherry in the small luminal loop. The RUSH *CD9-eGFP* and RUSH *CD81-eGFP* plasmids were obtained from this plasmid, replacing the *CD63* sequence by a synthetic sequence of human *CD9* or human *CD81* containing restriction sites allowing the later cloning of the *SBP-eGFP* or *mCherry* sequence from the RUSH-*CD63* plasmids and a linker (N-terminal sequences provided in Supplementary Table 1, restriction sites in bold, linker in italics) (gBlocks Gene Fragments purchased from Integrated DNA Technologies). The RUSH *CD63-YA-mCherry* plasmid was obtained by performing directed mutagenesis on the RUSH *CD63-mCherry* plasmid, using *CD63-YA* forward and reverse primers (Supplementary Table 1). The parental plasmid was digested with the DpnI enzyme (NEB). The *mCherry* was then replaced by *eGFP* to generate the RUSH *CD63-YA-eGFP* plasmid. The *Myr-Palm-mCherry* (available upon request to F. Perez) was generated by fusing the myristoylation–palmitoylation sequence and a linker in N-term of the *mCherry* sequence. Primers and original sequences used for cloning are indicated in Supplementary table 1. RUSH plasmids encoding WT and mutant CD9, CD63, CD81 are available upon request to C. Théry.

## EV isolation by differential ultracentrifugation

**EV isolation by differential ultracentrifugation**. FCS-EV depleted medium was first prepared by centrifuging DMEM with 20% FCS at 200K×g overnight with a

45Ti rotor (Beckman Coulter). Supernatant was then recovered by pipetting without taking the last 5 mL to avoid disturbing the pellet, and filtered through a 0.22 μm bottle filter (Millipore). Culture medium of (generally) 3 subconfluent (less than 80%) 150 cm dishes was changed, after a PBS wash, for FCS-EV depleted media diluted with DMEM to 10% FCS. Twenty-four or 16 h later, medium was collected for EV isolation and cells were trypsinized and counted (generally around $25 \times 10^6$ cells/dish). Three consecutive centrifugations of 20 min at 300 g at 4 °C were performed to remove any floating cells. Then serial centrifugations of the supernatant at 2K×g, 10K×g, and 200K×g at 4 °C were performed. 2K×g centrifugation was performed during 20 min, the pellet washed in 50 mL PBS and re-centrifuged for 20 min at 2K×g in an Eppendorf Centrifuge 5810. 10K×g and 200K×g centrifugations were performed in a 45Ti rotor (Beckman Coulter) during 40 min and 2 h, respectively. Pellets were resuspended in 6 mL PBS for a wash and centrifuged again at the same speed in a MLA80 rotor (Beckman Coulter) during 20 min and 1 h, respectively. Finally, the pellets were resuspended in 1 μL PBS/ $1 \times 10^6$ secreting cells.

**Nanoparticle tracking analysis (NTA).** NTA was performed using ZetaView PMX-120 (Particle Metrix) equipped with a 488 nm laser, at ×10 magnification, with software version 8.05.02. The instrument settings were 22 °C, gain of 26 and shutter of 70. Measurements were done at 11 different positions (2 cycles per position) and frame rate of 30 frames per second. Image evaluation was done on particles with minimum brightness: 20, minimum area: 10, maximum area: 500, maximum brightness: 255. Tracking radius2 was 100, and minimum tracelength: 7.

**Western blot.** Cell lysates (CL) for Western blot were obtained by incubating cell pellets at a concentration of $1 \times 10^6$ cells in 25 uL of lysis buffer (50 mM Tris, pH 7.5, 0.15 M NaCl, 1% Triton X-100) with 2% complete protease inhibitor (Roche) for 20 min on ice, followed by a 18,516×g centrifugation for 15 min at 4 °C to recover the supernatant. EVs from $20 \times 10^6$ cells (or specified number of particles) and CL from $0.2 \times 10^6$ cells were mixed with Laemmli sample buffer (BioRad), without reducing agent. After boiling for 5 min at 95 °C, samples were loaded on a 4–15% Mini-protean TGX stain-free gels (BioRad). Total proteins were imaged from the stain-free gels with the ChemiDoc Touch Imager (BioRad). Transfer was performed on Immuno-Blot PVDF membranes (BioRad), with the Trans-Blot Turbo Transfer System (BioRad) during 7 min. Blocking was performed during 30 min with Blocking Reagent (Roche) in TBS 0.1% Tween for most antibodies or with 5% milk in PBS 0.1% Tween for the anti-GFP antibody. Primary antibodies were incubated overnight at 4 °C and secondary antibodies during 1 h at room temperature (RT). Development was performed using either the BM Chemiluminescence Western blotting Substrate (POD) (Roche), Clarity Western ECL Substrate (BioRad) or the Immobilon Forte Western HRP substrate (Millipore), and the ChemiDoc Touch Imager (BioRad). Intensity of the bands was quantified using ImageJ.

**Immunoisolation of EVs by anti-CD63 and -CD9 for Western blot.** Concentrated conditioned medium (CCM) was prepared using serum-free DMEM incubated with secreting cells during 24 h. This medium was centrifuged at 300 g for 20 min to remove floating cells and filter-concentrated using Sartorius Vivaspin 100KDa molecular weight cut-off concentrators before incubation with the antibody-coupled beads. Two types of protocols were used: for Fig. 1c, Supplementary Fig. 1a, 100 μL of Pierce protein A magnetic beads (ThermoFisher) per sample were incubated with IgG, CD63, or CD9 antibodies overnight at 4 °C on agitation and washed three times with PBS 0.001% Tween. Processed CCM from $20 \times 10^6$ cells was incubated with antibody-coupled beads overnight in PBS 0.001% Tween on agitation at 4 °C. After 3 washes in PBS 0.001% Tween, pull-down samples were separated from the beads by heating at 95 °C during 5 min after addition of the loading buffer. For Supplementary Fig. 1b, Exosome Isolation kit beads (Miltenyi) were used following manufacturer's instructions. Briefly, CCM from $20 \times 10^6$ cells was incubated with 50 μL of anti-CD63 or anti-CD9 beads overnight at 4°C. Washes were performed on the columns and with the isolation buffer provided in the kit, and elution was performed with 25 μL Laemmli 1.5x (BioRad). Flow-through (FT) were recovered from the first supernatant pooled with 3 washes of beads, ultracentrifuged for 30 min at 200K×g using the TLA 100.3 rotor (Beckman Coulter). All the recovered materials of PD and FT were loaded on gels.

**Whole-mount and immuno electron microscopy on EVs.** Electron microscopy was performed on EV pellets resupended in PBS and stored at −80 °C that had never been thawed and re-frozen. EV suspension in PBS was deposited on formvar/carbon-coated copper/palladium grids and adsorbed for 20 min before uranyl/acetate contrasting and methyl-cellulose embedding for whole-mount analysis as described previously[37]. Double-staining with different antibodies were performed according to the Protein A-gold method[38] on EVs adsorbed to formvar/carbon-coated copper/palladium grids. CD9/CD63 double immunostaining was performed by successively incubating with rabbit anti-CD9 (Abcam ab236630 1/80) for 30 min, 10 nm protein-A-gold (CMC, Utrecht, The Netherlands) for 20 min, fixed for 5 min with 1% glutaraldehyde (Electron Microscopy Sciences), followed by mouse anti-CD63 (TS63 Diaclone 857.770.000 1/200) in PBS-BSA 1% for 30 min, rabbit

anti-mouse (Sigma SAB3701080 1/100) for 30 min, 15 nm protein-A-gold (CMC, Utrecht, The Netherlands) for 20 min, and fixed for 5 min with 1% (w/v) glutar-aldehyde in PBS. Subsequently, after a wash on 10 droplets of distilled water, grids were transferred to droplets of 0.4% (w/v) uranyl acetate (UA) staining and 1.8% (w/v) methylcellulose embedding solution. After 10 min of incubation, grids were picked up in a wire loop. Most of the excess of the viscous embedding solution was drained away with filter paper after which the grids with sections were air-dried forming a thin layer of embedding solution. Images were acquired with a digital camera Quemesa (EMSIS GmbH, Münster, Germany) mounted on a Tecnai Spirit transmission electron microscope (FEI Company) operated at 80 kV. To measure the size of the EVs with ImageJ software, their diameter was estimated from the mean between the height and the width of a rectangle that was drawn around each structure. A total of 593 EVs from HeLa cells in control situation, 574 EVS from cells treated with DMSO, 944 EVs from cells treated with GW4869, and 1112 EVs from cells treated with BafA1, pooled from 3 independent experiments, were quantified.

**Immunofluorescence on cells.** Cells were seeded on 12 mm diameter coverslips. Once they reached confluency (untransfected cells, Fig. 1e), 24 h after transfection with the RUSH-CD63-mCherry and -CD9-eGFP and 1 h of incubation with biotin (Fig. 2d), or 24 h after transfection with the RUSH-CD63-WT-eGFP and -CD63-YA-eGFP constructs in the presence of biotin (Supplementary Fig. 3a), they were fixed with 4% paraformaldehyde (PFA) (EMS) during 15 min at RT. Primary and secondary antibodies were successively incubated during 1 h each at RT in PBS containing 0.05% saponin and 0.2% BSA, except for anti-Rab7: cells were permeabilized in 0.3% Triton X-100 + 5% BSA (in PBS) for 15 min at RT, blocked in 5% BSA (in PBS) for 30 min at RT, primary antibody was incubated overnight at 4 °C and secondary antibody 1 h at RT, both in blocking solution. Coverslips were then mounted on slides with Fluoromount G (Invitrogen) with DAPI. Images were acquired on a Zeiss LSM 780 confocal microscope using a 63x objective with 1.46 size aperture, with the following acquisition parameters: frame rate 1 Hz, average per line 2, pixel size depending on the sample between 65 and 85 nm, pixel dwell 1.12 μsec, z-step 0.33 μm for stack imaging. At least 5 fields were captured to image a total of at least 10 cells per experiment (two independent experiments). Image analysis was performed with ImageJ. To focus on small compartments, a median filter of 1 pixel radius was first applied to remove noise, then a subtract background with a small rolling radius (between 10 and 50 pixels) was done to remove intensities from the large Golgi area. Then, to measure colocalization between two channels, JACoP plug-in[39] was used to calculate Mander's coefficients in each individual cell. Intensity thresholds to measure Mander's coefficients were set manually for each cell, according to the level of intensity of each channel. For colocalization between three channels in individual cells, a homemade macro was used to compute Mander's coefficients. First, 3D masks of Rab7-, eGFP-, and mCherry-positive compartments were performed with a user-defined intensity threshold after background subtraction as previously. Then, Mander's colocalization coefficients were calculated based on the percentage of overlapping volume for those masks.

**Live imaging of RUSH constructs.** Cells were transfected 24 h before imaging and washed with PBS before adding fresh complete culture medium 8 h later. To analyze steady-state distribution of the RUSH constructs, cells were transfected and cultured throughout in the presence of 40 μM biotin. For synchronized analysis of RUSH constructs, cells were transfected and cultured in complete medium until the time of biotin addition. Live imaging was performed with an Eclipse 80i microscope (Nikon) equipped with spinning disk confocal head (Yokogawa), using a 60x objective with 1.4 size aperture and a iXon Ultra897 camera (512 × 512 μm, pixel size 13 μm, Andor). 25mm-diameter coverslips with the transfected cells were put in a L-shape tubing Chamlide (Live Cell Instrument), filled with pre-warmed carbonate independent Leibovitz's medium (Invitrogen) with 1% FCS. Medium was replaced by pre-warmed Leibovitz' medium (Gibco) with 1% FCS and 40 μM biotin at time 0. For experiments with NH$_4$Cl, medium was replaced after 1 h of incubation with biotin for medium with biotin and 50 mM NH$_4$Cl. Movies were acquired using Metamorph software, imaging every 1 min for single color imaging, 2 min for two colors imaging, and an exposure time of 100 ms for each channel. 5 fields were imaged per replicate to image at least a total of 10 cells per replicate. For quantifications, automated image analysis was performed with ImageJ. We generated macros to quantify the kinetics of trafficking. Briefly, a first macro was used to count along time the number of small punctual fluorescent compartments (less than 15 pixels), using the "find maxima" function. This homemade macro was also used to measure the fluorescence intensity in the Golgi (defined by a threshold of 20 pixels) and in large compartments (threshold of 15 pixels) when the fluorescence in the Golgi decreased. Starting time for measurement of fluorescence in large compartments after exit from the Golgi was manually defined. Another macro was used for the co-localization analysis. Pearson's coefficients between the 2 markers were computed inside the cytoplasm at each time using Coloc2 plugin. A Pearson's coefficient of 1 indicates full colocalization. Medians of Pearson's coefficients calculated in at least 10 cells per experiment in 2–3 independent experiments are shown.

**Electron microscopy on cells**. Sample preparation, ultrathin cryosectioning, and immunolabelling were performed as already described[38]. In brief, HeLa cells were grown on culture dishes and fixed by the addition of freshly prepared 4% PFA in 0.1 M phosphate buffer (pH 7.4) to an equal volume of culture medium for 10 min, followed by postfixation with fresh 4% PFA overnight at 4 °C. After rinsing with PBS, the blocks were embedded in 12% gelatin, cryoprotected with 2.3 M sucrose, and frozen in liquid nitrogen. Ultrathin cryosections were cut on a Leica ultracut UC7 cryomicrotome and picked up in a freshly prepared 1:1 mixture of 2.3 M sucrose and 1.8% methylcellulose, thawed and collected on formvar-coated grids. After washing with PBS containing 0.02 M glycine, sections were incubated with primary antibodies and protein A-gold conjugates (PAG) (Utrecht University, The Netherlands). The following antibodies were used at the indicated dilutions: anti-GFP abcam ab290 1/800; anti mCherry GenTex GTX 128508 1/250. Sections were examined using a Tecnai Spirit electron microscope (FEI Company) equipped with a digital camera Quemesa (SIS), and iTEM 5.2 software (Olympus Soft Imaging Solutions GmbH). Quantitative IEM analysis was performed as described by Mayhew et al.[40] using ImageJ v1.53c. A total of four compartments, including MVBs, ER, Golgi, and PM, were selected to analyze the intracellular distribution of CD63 and CD9. The expected distribution was obtained by superimposing to pictures an array of points that was generated digitally and points (P) were counted in the selected compartments. Pictures were taken randomly from 2 independent experiments with the only criterion of a well-preserved morphology, and 7 different pictures per experiment were used for quantification. Gold particles on all sampled fields were also counted and named as observed gold particles (Ngo). For each compartment, the observed numbers of golds (Ngo) was compared with the expected numbers of golds (Nge, derived from the observed frequencies of point P). LD is calculated as the number of gold particles per test point (LD = Ngo/P). For each compartment, RLI = LD comp /LD cell. RLI = 1 indicates random labeling but RLI > 1 indicates when compartments are preferentially labeled[40]. By means of a two-sample Chi-squared ($\chi^2$) analysis with two columns (Ngo) and (Nge) and c compartments (arranged in rows), two distributions were compared, the total and partial $\chi^2$ values were calculated, and whether to accept or reject the null hypothesis was decided (of no difference between distributions) for $c - 1$ degrees of freedom. For any given compartment, the partial $\chi^2$ is calculated as (Ngo − Nge)²/Nge. If the observed and expected distributions are different, examining the partial $\chi^2$ values will identify those compartments that are mainly responsible for that difference. A convenient arbitrary cut-off is a partial $\chi^2$ value accounting for 10% or more of total $\chi^2$ [40]. Complete quantification table is shown in the source data file.

**Cell surface staining**. After 30 min, 1 h, 2 h of incubation with biotin, or no incubation (T = 0), or at steady-state (cells transfected already in the presence of biotin), cells were detached with 0.5 mM EDTA, washed and resuspended in cold PBS with 1% FCS, and placed on ice to stop the trafficking during 10 min. Cells were then incubated with the anti-GFP-AF647 for 40 min on ice and washed, before fixation with 2% PFA at RT during 10 min. After washes, cells were resuspended in PBS with 1% FCS and analyzed for GFP and AF647 fluorescence on a FACS Verse (BD), using the software BD FACSuite v1.0.6 and the following voltages for each laser: FSC: 72.8; SSC: 292.7, FITC: 281.7, APC: 356.2. Data were analyzed using FlowJo 10.6. Gating excluding debris was based on FCS/SSC and gating excluding doublets was based on FSC-A/FSC-H (Supplementary Fig. 5). Non-transfected and non-AF647-stained cells were used for gating the cells positive for GFP and AF647. No compensation was necessary to analyze GFP and AF647 signals obtained from two different lasers (FITC and APC). Data were not transformed. The increase of surface exposure at a given time point was calculated for cells in this gate by the formula: $(MFI_{AF647}/MFI_{GFP})tn - (MFI_{AF647}/MFI_{GFP})t0$.

**Antibody uptake**. After 2 h of incubation with biotin, cells were (1) washed with cold PBS with 1% FCS and placed on ice for 10 min, (2) incubated with anti-GFP-AF647 during 1 h on ice and washed, (3) placed into complete culture medium at 37 °C or 4 °C (negative control condition with no internalization) during 1 h, and (4) incubated for 5 min with trypsin (stripping) or 0.5 mM EDTA (no stripping) at 37 °C. Detached cells were then fixed with 2% PFA for 10 min at RT, washed, and analyzed with the FACS Verse using the same settings as for the cell surface staining described above. Data were analyzed using FlowJo 10.6. Gating of the cells was performed as described in the previous section. Not transfected cells were used as a negative control for gating of the GFP positive cells. The percentage of GFP that has been internalized during 1 h at 37 °C is calculated from the ratio between the stripped and non-stripped conditions of AF647/GFP signal (in the GFP-positive gate), after subtracting the background internalization signal calculated similarly at 4 °C, by the following formula: $(MFI_{stripping}x100)/MFI_{no\ stripping})_{37°C} - (MFI_{stripping}x100)/MFI_{no\ stripping})_{4°C}$.

**Immunoisolation of RUSH CD63- and CD9-eGFP EVs for Western blot and proteomics**. A total of $2.5 \times 10^6$ cells per 10 cm dish were plated on the day before transfection. Twenty-four hours after transfection, DMEM without FCS with 40 nM biotin was incubated during 3 h or 24 h with cells transfected with the RUSH plasmids CD63- or CD9-eGFP, or non-transfected cells as control. Harvested medium from 12 plates per transfection or 18 plates of non-transfected cells was centrifuged for 20 min at 300 g and concentrated with Centricon plus-70 filters

(Millipore) with a 100KDa molecular weight cut-off. The CCM concentrated to 500 uL was then submitted to size exclusion chromatography using qEV 70 nm columns (Izon). Fractions 7 to 12 were recovered and pooled according to manufacturer's instructions. The pooled fractions were finally concentrated using 500uL 100 kDa molecular weight cut-off concentrators (Sartorius). The percentage of transfected cells was determined by flow cytometry and the number of recovered EVs was measured by NTA. For 3 h samples, EVs from $20 \times 10^6$ GFP-positive cells were incubated with the beads. An equivalent number of total particles from the non-transfected control sample was used. For 24 h samples, a total of $60 \times 10^8$ particles was incubated in duplicate with the beads. The protein G beads were previously hybridized with the anti-GFP overnight and cross-linked using BS[3] (ThermoFisher) for 1 h at 4 °C and finally washed with PBS 0.001% Tween. Incubation with the beads was performed immediately after EV isolation, in PBS 0.001% Tween overnight at 4 °C in rotation. One PD duplicate was used for Western blot and the other one for mass-spectrometry. For Western blot (Fig. 5b), the beads were then washed three times with PBS 0.001% Tween with 300 mM NaCl and three times with PBS. The flow through pooled with the first three washes was concentrated using 500uL 100 kDa molecular weight cut-off concentrators. All the sample volume recovered from the PD and the FT was loaded on the gel. For mass-spectrometry, immuno-isolated vesicles on beads were eluted with 100 µL 80/20 MeCN/H2O + 0.1 % TFA. Dry pellets were solubilized and reduced in 20 µL 8 M urea, 200 mM ammonium bicarbonate, 5 mM dithiothreitol, pH 8 with vortexing at 37 °C for 1 h. After cooling to room temperature, cysteines were alkylated by adding 10 mM iodoacetamide for 30 min in the dark. After diluting to 1 M urea with 200 mM ammonium bicarbonate pH 8.0, samples were trypsine/LysC (1 µg, Promega) digested in a total volume of 200 µL with vortexing at 37 °C overnight. Samples were then loaded onto home-made C18 StageTips for desalting. Peptides were eluted using 40/60 MeCN/H2O + 0.1% formic acid, vacuum concentrated to dryness, and reconstituted in injection buffer (2% MeCN/0.3% TFA) before nano-LC-MS/MS analysis. Four independent biological replicates of each sample were analyzed.

**Mass spectrometry analysis**. Liquid chromatography (LC) was performed with an RSLCnano system (Ultimate 3000, Thermo Scientific) coupled online to a Q Exactive HF-X mass spectrometer (MS) with a Nanospay Flex ion source (Thermo Scientific) peptides were first trapped on a C18 column (75 µm inner diameter × 2 cm; nanoViper Acclaim PepMapTM 100, Thermo Scientific) with buffer A (2/98 MeCN/H2O in 0.1% formic acid) at a flow rate of 2.5 µL/min over 4 min. Separation was then performed on a 50 cm × 75 µm C18 column (nanoViper Acclaim PepMapTM RSLC, 2 µm, 100 Å, Thermo Scientific) regulated to a temperature of 50 °C with a linear gradient of 2% to 30% buffer B (100% MeCN in 0.1% formic acid) at a flow rate of 300 nL/min over 91 min. MS full scans were performed in the ultrahigh-field Orbitrap mass analyzer in ranges m/z 375–1500 with a resolution of 120,000 at $m/z$ 200. The top 20 intense ions were subjected to Orbitrap for readout of MS fragments from the high-energy collision dissociation cell (HCD), analyzed at 15,000 resolution, with the intensity threshold kept at $1.3 \times 10^5$. We selected ions with charge state from 2+ to 6+ for screening. Normalized collision energy (NCE) was set at 27 and the dynamic exclusion of 40 s. For identification, the data were searched against the Homo sapiens (UP000005640) UniProt database using Sequest-HT through Proteome Discoverer (version 2.4). Enzyme specificity was set to trypsin and a maximum of two-missed cleavage sites was allowed. Oxidized methionine, carbamidomethyl cysteines, and N-terminal acetylation were set as variable modifications. Maximum allowed mass deviation was set to 10 ppm for monoisotopic precursor ions and 0.02 Da for MS/MS peaks. The resulting files were further processed using myProMS[41] v3.9. FDR calculation used Percolator[42] and was set to 1% at the peptide level for the whole study. The label-free quantification was performed by peptide Extracted Ion Chromatograms (XICs), computed with MassChroQ version 2.2.1[43]. XICs from proteotypic peptides between compared conditions (TopN matching) with missed cleavages were used. Median and scale normalization was applied on the total signal to correct the XICs for each biological replicate ($N = 4$). To estimate the significance of the change in protein abundance, a linear model (adjusted on peptides and biological replicates) based on a two-tailed $T$-test was performed and $p$-values were adjusted with the Benjamini–Hochberg FDR procedure. In order to eliminate non-specifically isolated proteins, only proteins with at least two peptides across 2 biological replicates of an experimental condition and that showed a log2(fold change) > 0 in the samples containing GFP versus NT samples were selected. Proteins recovered in more than 200/411 IP analyses according to the contaminant repository for affinity purification (the CRAPome[44]), hence the contaminant commonly isolated or identified non-specifically by this technic, were removed from further analysis. Proteins selected according these two criteria (log2(fold change) GFP/NT > 0 and not contaminant according to the CRAPome) are highlighted in green in Supplementary data File 1, and those showing at least 2 peptides in at least 2 replicates of the CD9-eGFP and/or the CD63-eGFP samples were used for the quantitative analysis (Supplementary data file 2 and Fig. 6a). Proteins identified by at least 2 peptides identical in both CD9-eGFP and CD63-eGFP samples were considered enriched in one sample compared to the other if they showed a log2(fold change) $\geq 1$ or $\leq -1$ and an adjusted $p$-value of less than 0.05. Proteins with a log2(fold change) between −1 and 1 were considered as common to two samples. Proteins displaying peptides exclusively in CD63-eGFP or

in CD9-eGFP and with at least 2 peptides in 2 replicates were listed as unique to the corresponding sample.

GO term analysis was performed using the FunRich software[17,18] using the HeLa Spatial Proteome database[16]. The hypergeometric uncorrected *p*-value reflecting if there is an enrichment of proteins from one cellular component compared to the reference database, and the percentage of proteins compared to the total number of proteins in the list were calculated by the software for each cellular component.

**Statistical analyses**. Following the recommendation of D.L.Vaux[45], for each experiment where number of biological replicates were 2 or 3, we displayed the results in a transparent manner, showing each individual biological replicate as a dot, so the readers could interpret the data for themselves. We (as suggested by D. L. Vaux) considered that same trends of results obtained independently 2–3 times were as informative as statistical tests to evaluate reproducibility of the experiments.

Nonetheless, we also performed statistical analyses with GraphPad Prism version 8.0.2 (GraphPad software, California USA), by one-way ANOVA followed by multiple comparison Tukey's test (Figs. 2d, 4b, 5b, 7a, Supplementary Fig. 3c, Supplementary Fig. 4a,b,c,d,e), by two-tailed paired *t*-test (Fig. 5a,d), two-tailed unpaired t-test (Supplementary Fig. 3a) or one-sample two-tailed *t* test (Figs. 5c and 7c, e).

## Data availability

The mass spectrometry proteomics raw data have been deposited to the ProteomeXchange Consortium via the PRIDE[46] partner repository with the dataset identifier PXD021515. Methods of EV isolation and analysis were submitted to the EV-TRACK platform[47] (EV-TRACK ID: EV210105 [https://evtrack.org/search.php]). All images (fluorescence and electron microscopy) have been deposited in BioImage via accession number S-BIAD130. In-house generated ImageJ macros have been deposited to Zenodo with the following references: 4767826 [https://doi.org/10.5281/zenodo.4767826]. This concerns the following figures: Figs. 1a–c, 2b–d, 3c, 4a–b, 5a–d, 7a–e, 8, Supplementary Figs. 1a–b, 2a–b, 3a–c, 4a–e. The remaining data are available in the Article and Supplementary Information. Source data are provided with this paper.

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

## Acknowledgements

We thank for fruitful discussions several team members, especially Drs Mercedes Tkach, Eleonora Grisard, Lorena Martin-Jaular, Jason Ecard, and for helpful discussions and tools, Dr Aurélien Dauphin, Institut Curie, Paris, Dr P. Zimmermann, KU Leuven, Belgium and CRCM Marseille, France, Dr G. van Niel, IPN Paris, France, Dr P Benaroch, Institut Curie, Paris, France, Dr Suresh Mathivanan, LaTrobe University, Melbourne, Australia. The H4A3 monoclonal antibody (anti-human LAMP1), developed by August, J.T. and Hildreth, J.E.K. from Johns Hopkins University School of Medicine, was obtained from the Developmental Studies Hybridoma Bank, created by the NICHD of the NIH, and maintained at The University of Iowa, Department of Biology, Iowa City, IA 52242. This work was funded by INSERM, CNRS, Institut Curie, French IdEx and LabEx (ANR-10-INSB-04, ANR-10-IDEX-0001-02 PSL, ANR-10-LABX-0038, ANR-11-LABX-0043, ANR-18-IDEX-0001 Université de Paris), grants from french ANR (ANR-18-CE13-0017-03; ANR-18-CE15-0008-01; ANR-18-CE16-0022-02), INCa (INCA-11548), Fondation ARC (PGA1 RF20180206962), FRM (FDT201904007945, EQU201903007925 and DGE20121125630), Cancéropôle Île-de-France (2013-2-EML-02-ICR-1), USA NIDA (DA040385), and a Long-Term EMBO Fellowship (ALTF 607-2015) co-funded by the European Commission FP7 (Marie Curie Actions, LTFCOFUND2013, GA-2013-609409) to J.I.V. We also acknowledge the following platforms of Institut Curie: PICT-IBiSA, member of the France-BioImaging national research infrastructure (ANR-10-INBS-04) for fluorescence and electron microscopy, flow cytometry for assistance in data acquisition, genomic for authentication of HeLa cells by STR analysis.

## Author contributions

M. Mathieu and C.T. designed the study, interpreted the data, and wrote the article. M. Mathieu, N.N., M.J., J.I.V., F.D., and D. Lankar performed experiments. M. Mathieu, N.N., M. Maurin, and M.J. analyzed the data. M. Mathieu, N.N., F.V., R.P., E.R., and G.B. generated plasmids or KO cells. G.B. and F.P. designed and supervised the RUSH studies. D. Loew designed and supervised the proteomic study. E.R. interpreted data. All authors read and corrected the article.

## Competing interests

The authors declare no competing interests.
