## [Peer Review File · Nature Communications]

REVIEWER COMMENTS

Reviewer #1 (Remarks to the Author):

Mathieu et al. aim to gain insights in the exosomal or ectosomal origin of extracellular vesicles (EVs) and their respective cargo using HeLa cells by 1) implementing the RUSH system to study the intracellular trafficking of CD9 versus CD63, 2) performing proteomic analysis of CD9 versus CD63 bearing EVs, and 3) using bafilomycinA1 to neutralize endosomal pH. This work contributes technical advances to facilitate the study of the release of EV subsets (including the RUSH system and bafilomycinA1) and biological advances in markers that can distinguish EV subsets from different origin, at least in HeLa cells. The authors reveal that both CD9 and CD63 are released in small ectosomes which are further characterized by the presence BSG and SLC3A2 and insensitivity to bafilomycinA1. By contrast, CD63 is released in exosomes that are further characterized by the presence of LAMP1/2 and sensitivity to bafilomycinA1. Furthermore the authors demonstrate that GW4869 inhibitor, an inhibitor commonly used to block exosome release, does not only affect exosome biogenesis, but also release of other EV subsets. Both the technological and biological advances presented in this study are highly relevant to the EV research field.

Major comments

-“By centrifuging further at 200K we recovered some more CD9 and CD63 in material that had not been pelleted at 100K”. The authors implement differential ultracentrifugation to enrich for small EVs. Typically a 100K pelleting step is included, as also previously implemented by the authors, to pellet small EVs. Although pelleting at 200K will recover more EVs, this step will also likely recover other particles. The authors should perform a full characterization of the 200K pellet according to the MISEV2018 guidelines including western blot analysis of membrane proteins and cytosolic proteins, non-EV associated proteins (expected contaminants), electron microscopy and quantitative analysis. Furthermore, the authors focus their study on small EVs, but what are small EVs? The only size measurement made is presented in Figure 7 revealing a broad size distribution from 70-300 nm using NTA (with a presumably lower cut-off of 70 nm). Also here electron microscopy is required to understand the size the small EVs that are studied and to properly define the nomenclature of the small EV studied in this manuscript.

-In general, small EVs obtained from CD63, CD63-YA and CD9 transfected Hela cells or Hela cells treated with bafilomycinA1 or GW4869 should be studied including electron microscopy, NTA, western blot for non-EV associated proteins.

-How was the treatment condition for bafilomycinA1 optimized (100 nM)? This treatment dose is high compared to what is described/used in literature (0.1 to 1 nM). What is the effect on cell toxicity? What is the cell viability and cell morphology? Is the treatment reversible?

-In Figure 5A a substantial signal is identified for CD9 in the 10K pellet, this is sharp contrast to Figure 1. How do the authors explain this? If CD9 is also abundant in the 10K pellet, do the authors observe any effects on the release of large EVs using the RUSH system and bafilomycinA1 or GW4869?

-Pearson's coefficients are established for a period of 100 to 120 min. Proteomics is performed at 3h and 24h. However, what is the localization of CD63 and CD9 from the start of the experiment up to 3h and 24h? What happens in this time frame and how does this compare to the image analysis?

-All the observations have been made using one cell line. The authors should confirm key findings using additional (cancer) cell lines.

Minor comments

- Page 2: we adapted CD63 and CD9 “to”
- Page 3: was observed “in” a steady state
- Submit experimental parameters to EV-TRACK
- MIFlowCyt criteria are not fulfilled.
- What is the origin of the HeLa cells? Is a STR profile available?
- Why use serum-free DMEM for immunoprecipitation, while in other experiments EV-depleted FBS was used? Please specify.
- Live cell imaging + image analysis: Image acquisition parameters are not included (frame rate, pixel dwell time, pixel size, resolution, ...)

Reviewer #2 (Remarks to the Author):

This manuscript addresses whether CD9 and CD63 are markers of different small EV populations, specifically of small ectosomes (CD9) or exosomes (CD63). This would make sense given their steady state localization respectively to the plasma membrane (CD9) or endosomes (CD63). The authors used the RUSH system to examine the dynamic trafficking of CD9 and CD63 to internal endosomes and the plasma membrane. At first the CD9 and CD63 molecules traffic to the plasma membrane and then segregate to their respective locations. At 3 h there is more mixing of the two markers whereas by 24 hours, they seem to be more localized to their steady state localization. The authors used a few ways to look at which vesicles were ectosomes versus exosomes coming from the cells. Use of Bafilomycin to neutralize acidic organelles (late endosomes, lysosomes) increased the release of CD63 and some endosomal and lysosomal markers, along with syntenin. These data suggest (but do not prove) that neutralization of lysosomes prevents their fusion with MVBs, allowing more MVBs to fuse with the plasma membrane than otherwise would. An orthogonal approach is using a mutant of CD63 that doesn't localize to lysosomes. This leads to localization to both the plasma membrane and small internal vesicular organelles and also enhances secretion of CD63 in EVs. This is taken as proof that more small EVs are released from HeLa cells as ectosomes than as exosomes – although it seems possible that CD63 itself affects biogenesis (this is actually shown) no matter where it is, but that many of the exosomes formed in MVBs are degraded in lysosomes. So that result may not be generalizable to all EVs formed under steady state. Overall, this is a very interesting paper that addresses an issue in the field that is quite important. That is, can small EVs be distinguished as ectosomes or exosomes based on protein cargo markers and what proportion of small EVs are in those categories? As it was all done in one cell line, it is a bit difficult to be conclusive about what proportion of small EVs released from cells are exosomes versus ectosomes. However, the authors do convincingly show that CD63 is an excellent marker of exosomes whereas CD9 is much more likely to mark small ectosomes. There is much to like about this study, but in some places the data are quite thin that underlie the conclusions. In addition, some of the data may be over- or mis-interpreted. Specific comments are:

1. There is missing statistical analysis for most of the figures and many of the experiments do not have adequate replicates (where noted). It is hard to see how one can make conclusions without enough replicates and statistical tests. All quantitations (e.g. the Western blot graphs) should have statistics and adequate replicates (e.g. cell stainings). In some cases, there is not a notation of how many independent replicates and/or cells/image fields there are. Typically for cell images one would want ~10-20 fields per biological replicate and 3 biological replicates.

2. While replicates and image fields are mentioned in some figure legends, they are not in others (E.g. Supp Fig 5)

3. Supp Fig 1. This is a very nice experiment, combining the conditioned media of KO cells. However, the interpretation does not seem totally correct as there is some pull-down of CD63 from the combined KO CM in the CD9 IP. Also, there is no quantitation of the results. The point about whether CD9 and CD63 are really present on the same EVs could be addressed by immunoelectron microscopy.

4. Supp Fig 3 and Fig 3, what are the small compartments that the CD63-YA-GFP goes to? Some labeling with early, late, and recycling endosome compartments are in order (e.g. Rab5, Rab7, Rab11...) as well as colocalization with lysosomal markers (LAMP1)

5. Supp Fig 4: It looks like CD63-YA-GFP and CD63-GFP expression both increase the total number of small EVs released from cells at 24 h in comparison to nontransfected cells and/or CD9-GFP cells. It seems like these data are in contradiction to the conclusion that more ectosomes are released from HeLa Cells than exosomes. It would seem to me that the number of EVs is a more reliable readout of EV biogenesis than CD63 levels. It also seems like all of the constructs are affecting the process they are being used to measure. How do the authors interpret these data and how do the levels of expressed proteins compare to each other? I don't see a Western blot directly comparing the levels of CD63-YA-GFP, CD63-GFP, and CD9-GFP in transfected cells to each other using a GFP antibody.

6. Fig 5C: The band intensity for the GFP-YA-CD63 looks to be increased substantially by Bafilomycin, unlike what is stated in the text and what is shown in the bar graph. Why is that? Is it not a representative blot or was there a problem in quantitation of the blot?

7. Related to Fig 6 - Bottom of page 6 - This sentence is very confusing: "This suggests that a common mechanism of release (or subcellular origin) of CD63 EVs and a fraction of CD9 EVs and another mechanism specific of CD63 EVs are both equally important early after CD63 and CD9 synthesis." The sentence is convoluted and hard to understand. I also think that the authors may be reading too much into their data – it seems pretty clear that CD63 is mostly in endosomes and CD9 is mostly at the plasma membrane and the EVs reflect that. There is some overlap due to natural trafficking between those two compartments.

8. Fig 6B. The statistics that are done are a little hard to understand – the legend says it is in relation to the representation in the database – what does that mean with respect to this study? Does that mean that there are not comparisons within the study? For example, there are a lot of mitochondrial proteins in the 3h CD9+ sample but not at 24 h. Is that meaningful?

9. Fig 6 (potential Discussion points) – could the authors comment on the relationship of the Endosome and Lysosome categories to Late Endosomal proteins, such as would be expected to represent MVBs? Also, how does the 24 h sample relate to steady state samples?

10. Discussion – there are some confusing sentences, like:
"Concerning the PM-EV markers: SLC3A2 is the heavy chain of various heterodimeric amino-acid transporters, and one of its possible partners, SLC7A5 (also named E16 or LAT-1)²³, was less specific of the CD9-EVs, but not enriched in the CD63-EVs, thus confirming an ectosomal release."

11. It is a bit surprising that some EV characterization, as outlined in MISEV2018 and the earlier MISEV, are missing from this study – such as TEM and checking negative markers on Western blot.

Reviewer #3 (Remarks to the Author):

Technical review of the used LC-MS of captured extracellular vesicles.

Matthieu et al. describe the use of an elegant system making use of the RUSH system in which they 'stall' release of their proteins of interest from e.g. the ER to image the synchronized trafficking of proteins used as specific EV markers. The study is well described and addresses an important question in the field of EV biogenesis. The aim of this review is to assess the technical quality of the performed MS-based proteomics experiments to identify EV subtype-specific proteomes.

In their approach the authors used their adapted RUSH system to immunocapture CD63- and CD9-eGFP EVs, using an anti-GFP antibody. This is an elegant combination that allows for the enrichment of freshly secreted EVs using the same antibody for both conditions, with a non-transfected control. The authors performed all experiments 4-fold and performed the statistical analysis with FDR correction. Overall the data is robust and well presented. A few small comments below.

1. The authors make some claims on the end of page 6 and start of page 7, based on the observation of a slight increase of common (CD63/CD9) proteins (44% to 48%) and a decrease of CD63 specific proteins (37% to 32%) from 3hrs to 24hrs. They state this observation indicates different mechanisms of release for the different populations and increased dominance of one over time. Given the very small differences, it's questionable these observations are statistically significant and importantly, an observation in a single cell line (Hela) is not sufficient for such general statements. If the authors really believe this, they should show evidence for this in multiple cell lines and perform the proper statistical analysis.
2. Supp figure 3B is very busy and difficult to interpret at first glance. The quality of the representation can be improved.
3. In the description of the MS analysis on page 16 one sentence is a bit poor: "The top 20 intense ions were subjected to Orbitrap for further fragmentation via high energy collision dissociation (HCD) activation" Ions are not fragmented in the Orbitrap, but used for readout of MS fragments from the HCD collision cell. "and a resolution of 15 000" the authors probably mean these fragments are analyzed at 15 000 resolution. " 1.3×10^5 " should be 10^5 .

Point-by-point response to reviewers comments on manuscript NCOMMS-20-38099

REVIEWER COMMENTS

Reviewer #1 (Remarks to the Author):

Mathieu et al. aim to gain insights in the exosomal or ectosomal origin of extracellular vesicles (EVs) and their respective cargo using HeLa cells by 1) implementing the RUSH system to study the intracellular trafficking of CD9 versus CD63, 2) performing proteomic analysis of CD9 versus CD63 bearing EVs, and 3) using bafilomycinA1 to neutralize endosomal pH. This work contributes technical advances to facilitate the study of the release of EV subsets (including the RUSH system and bafilomycinA1) and biological advances in markers that can distinguish EV subsets from different origin, at least in HeLa cells. The authors reveal that both CD9 and CD63 are released in small ectosomes which are further characterized by the presence BSG and SLC3A2 and insensitivity to bafilomycinA1. By contrast, CD63 is released in exosomes that are further characterized by the presence of LAMP1/2 and sensitivity to bafilomycinA1. Furthermore the authors demonstrate that GW4869 inhibitor, an inhibitor commonly used to block exosome release, does not only affect exosome biogenesis, but also release of other EV subsets. Both the technological and biological advances presented in this study are highly relevant to the EV research field.

We thank this reviewer for his/her positive evaluation of our work.

Major comments

-“By centrifuging further at 200K we recovered some more CD9 and CD63 in material that had not been pelleted at 100K”. The authors implement differential ultracentrifugation to

enrich for small EVs. Typically a 100K pelleting step is included, as also previously implemented by the authors, to pellet small EVs. Although pelleting at 200K will recover more EVs, this step will also likely recover other particles. The authors should perform a full characterization of the 200K pellet according to the MISEV2018 guidelines including western blot analysis of membrane proteins and cytosolic proteins, non-EV associated proteins (expected contaminants), electron microscopy and quantitative analysis.

Furthermore, the authors focus their study on small EVs, but what are small EVs? The only size measurement made is presented in Figure 7 revealing a broad size distribution from 70-300 nm using NTA (with a presumably lower cut-off of 70 nm). Also here electron microscopy is required to understand the size the small EVs that are studied and to properly define the nomenclature of the small EV studied in this manuscript.

Our study was designed at following specifically 2 particular transmembrane EV markers, CD9 and CD63, and determining the nature of the EVs in which they are released, rather than at characterizing the bulk EV preparations used to follow them. Since our first observation (figure 1a) was that hardly any material was recovered in the first steps of centrifugation, which in other cells pellet large- and medium-sized EVs (according to our previous work: *Kowal, PNAS 2016*), we used the term “small EVs” for the high-speed pellets enriched in CD9 and CD63. However, we agree that reliable use of this term was not properly demonstrated by size characterizations. We have thus, in this revised version, answered the reviewers’ and editor’s requests and performed additional characterizations of the bulk EVs isolated by differential ultracentrifugation (2K, 10K and 200K) from the conditioned medium of HeLa cells, as recommended by the MISEV2018 guidelines. In control (new figures 1a,b,d), and following drug treatment conditions (figures 7c,d,e), we display Western blots showing the 3 categories of proteins required to demonstrate the presence of EVs: membrane-associated (CD63, CD9 and new CD81), cytosolic (syntenin-1), and 2 complementary controls: the ER protein calnexin which detects presence of EVs from other origins than the endosomes or plasma membrane, and the mainly non-EV associated AChE which controls for presence of non-EV material, as we showed in a previous study (*Liao, Martin-Jaular et al, J Extracell Vesicles 2019: 8, 1628592*).

We also analyzed the 200K pellets by NTA and by electron microscopy (new figure 1b, new figure 7d). Both show that EVs are smaller than 300nm in diameter, with a majority of vesicles of around 50 nm detected by EM but not by NTA, another important population of around 120nm, and less abundant EVs of 150-300nm. These features are now described in the first paragraph of results, and confirm that the bulk preparation of HeLa-derived EVs contain mainly EVs smaller than 250nm, called here small EVs.

Note that presence of AChE indeed suggests some contamination by serum-derived components of the EV preparations recovered by 200K ultracentrifugation. However, since we showed previously that AChE is mostly on very small particles that are not labeled by anti-CD63 (*Liao, Martin-Jaular, JEV 2019*), and we did not identify AChE in the proteomic analysis of the RUSH-CD63 nor -CD9-expressing EVs (suppl Table 2), its presence does not affect the specific analysis of the cell-derived CD9- and CD63-containing EVs, which is the purpose of our work.

-In general, small EVs obtained from CD63, CD63-YA and CD9 transfected HeLa cells or HeLa cells treated with bafilomycinA1 or GW4869 should be studied including electron microscopy, NTA, western blot for non-EV associated proteins.

For EVs recovered from HeLa cells transfected with the 3 RUSH constructs, they were already characterized by NTA of the bulk EVs (shown in former suppl figure 4A, now suppl Fig 4b) and proteomic analysis of the EVs expressing the specific construct (figure 6, former suppl Figure 4B, now figure 6a). This proteomic analysis provides all relevant information on the CD63- and CD9-positive EV components and the potential contamination by non-EV associated proteins. We have now included an additional panel showing the mean size of bulk EVs, measured by NTA (new suppl figure 4c): no difference is observed between the different conditions. We do not think that an EM picture of these bulk preparations, which contain a mixture of EVs from cells expressing or not the RUSH-constructs, would provide any additional information.

For EVs recovered upon cell treatments with the drugs, we have now followed this reviewer suggestion to more extensively characterize them. We prepared new samples to perform electron microscopy (new figure 7d), NTA on both the 200K (previously shown in figure 7A, now 7c), and now on the 10K pellets, and Western blots with additional negative controls: calnexin and AChE (bottom panels in former figure 7C, now 7e). The other transmembrane and cytosolic markers suggested by MISEV2018 to characterize EVs were already present in the WB of figure 7C, now 7e.

-How was the treatment condition for bafilomycinA1 optimized (100 nm)? This treatment dose is high compared to what is described/used in literature (0.1 to 1 nM). What is the effect on cell toxicity? What is the cell viability and cell morphology? Is the treatment reversible?

For BafA1 treatment, we used the dose that was already described to affect EV secretion by HeLa cells in Edgar et al. eLife 2016. It does not affect cell viability compared to DMSO treatment as assessed by trypan blue staining (new figure 7b). We are not sure why the reviewer asks about reversibility of the BafA1 effect. We only used it for a 16h time-frame to follow its immediate effect on EV release, and we did not test other time-points or drug removal conditions.

-In Figure 5A a substantial signal is identified for CD9 in the 10K pellet, this is sharp contrast to Figure 1. How do the authors explain this? If CD9 is also abundant in the 10K pellet, do the authors observe any effects on the release of large EVs using the RUSH system and bafilomycinA1 or GW4869?

The signal of CD9 observed on Western blots of the 10K pellet varies slightly in different biological replicates, but it is generally detectable, upon sufficient development of the luminescence signal. New figure 1a shows some CD9 signal in the 10K pellet.

To illustrate this variability, and answer this reviewer's question about effect of the drugs on large EV release, we provide new panels in new figure 7c (NTA) and 7e (WB) with quantification of the effect of BafA1 and GW4869 treatments on 10K pellets in 3 independent biological replicates. The number of particles and WB signals are not significantly increased by BafA1, nor by GW4869, although for the latter the number of EVs recovered in the 10K pellets is systematically increased, as previously proposed by others (Menck et al, J Extracell Vesicles 2017).

The RUSH system does not induce release of large EVs, as shown by new suppl figure 4c (size of total EVs from the CM of RUSH-construct expressing cells, measured by NTA).

-Pearson's coefficients are established for a period of 100 to 120 min. Proteomics is performed at 3h and 24h. However, what is the localization of CD63 and CD9 from the start of the experiment up to 3h and 24h? What happens in this time frame and how does this compare to the image analysis?

To answer this question, we have performed new time-course experiments shown in new suppl figure 4e. Cells were co-transfected with the CD63-Cherry and CD9-GFP RUSH constructs, and fixed after 30min, 2h, 3h or 24h of incubation with biotin. Co-localization of the markers was measured by Pearson's correlation analysis: it shows a similar level of colocalization at 2h (the time-point of all live video-analyses) and 3h (the first time point used for EV isolation for the proteomic analysis), and a lower co-localization at 24h (the second time point of the proteomic analysis). This analysis confirms the relevance of the time-points chosen for the proteomic analysis.

-All the observations have been made using one cell line. The authors should confirm key findings using additional (cancer) cell lines.

We agree with this reviewer that it would now be very interesting to determine whether our results in HeLa cells apply to other cell types, and especially if other cells may release more exosomes than ectosomes, differently from HeLa. We hope that the tools we describe here will help us and other groups evaluate this question, and we had in fact already included a sentence to this purpose in the last paragraph of discussion, p.11. Following the editors' specific comment, we have not included such analyses in this revised version.

Minor comments

-Page 2: we adapted CD63 and CD9 "to"
Corrected as proposed

-Page 3: was observed "in" a steady state
Sorry for the typo, "was observed a steady-state" is now corrected to "was observed at steady-state"

-Submit experimental parameters to EV-TRACK

We have submitted our experimental parameters to EV-TRACK. Reviewers and editor may access and check our submission via the following URL: <http://evtrack.org/review.php>. Please use the EV-TRACK ID (EV210105) and the last name of the first author (Mathieu) to access our submission.

-MIFlowCyt criteria are not fulfilled.

To follow MIFlowCyt guidelines (checklist of Cytometry Part A, 2010, 77A:813), we have included in the M&M (paragraphs on Surface staining and Antibody uptake) some missing information on experimental settings. Given the very basic and simple gating performed to exclude only debris and doublets, we do not think that showing it as supplementary figure is useful, but we can provide such figure if editor and reviewers find it necessary.

-What is the origin of the HeLa cells? Is a STR profile available?

We performed in 2018 a STR profile on the HeLa cells, which had been used in the lab for more than 2 decades. We analysed 16 markers, for 9 of which ATCC data on HeLa cells were available, and we observed 100% identity between our cells and the ATCC data, thus

validating the cells used here as HeLa (see results below, for reviewer). This information has been included in the M&M section p.11.

Figure for reviewer: STR analysis of the HeLa cells used in this study.

STR34- Thery- March2018 Hela CT			data ATCC HeLa (ATCC® CCL-2™)		
Marker	Allele 1	Allele 2	Marker	Allele 1	Allele 2
D3S1358	15	18	D3S1358		
TH01	7		TH01	7	
D21S11	27	28	D21S11		
D18S51	16		D18S51		
Penta E	7	17	Penta E		
D5S818	11	12	D5S818	11	12
D13S317	12	13,3	D13S317	12	13,3
D7S820	8	12	D7S820	8	12
D16S539	9	10	D16S539	9	10
CSF1PO	9	10	CSF1PO	9	10
Penta D	8	15	Penta D		
AMEL	x		AMEL	X	
vWA	16	18	vWA	16	18
D8S1179	12	13	D8S1179		
TPOX	8	12	TPOX	8	12
FGA	18	21	FGA		

-Why use serum- free DMEM for immunoprecipitation, while in other experiments EV-depleted FBS was used? Please specify.

For EV isolation techniques that required pre-concentration of the conditioned medium (such as immunoprecipitation), we used serum-free medium to limit clogging by excess of serum-derived proteins of the filter-concentrator columns. Furthermore, for immunoprecipitation of CD63- or CD9-EVs, we preferred to use filter-concentration rather than ultracentrifugation, to avoid potential artificial aggregation of EVs, which has been observed by others on some biofluids like plasma (Linares 2015 ##26700615). These specific technical issues are now specified in the corresponding paragraph of results, p.3.

-Live cell imaging + image analysis: Image acquisition parameters are not included (frame rate, pixel dwell time, pixel size, resolution, ...)
This information is now included in the M&M paragraphs on “Immunofluorescence on cells” and “Live Imaging of RUSH constructs”, p15.

Reviewer #2 (Remarks to the Author):

This manuscript addresses whether CD9 and CD63 are markers of different small EV populations, specifically of small ectosomes (CD9) or exosomes (CD63). This would make sense given their steady state localization respectively to the plasma membrane (CD9) or endosomes (CD63). The authors used the RUSH system to examine the dynamic trafficking of CD9 and CD63 to internal endosomes and the plasma membrane. At first the CD9 and CD63 molecules traffic to the plasma membrane and then segregate to their respective locations. At 3 h there is more mixing of the two markers whereas by 24 hours, they seem to be more localized to their steady state localization. The authors used a few ways to look at which vesicles were ectosomes versus exosomes coming from the cells. Use of Bafilomycin to neutralize acidic organelles (late endosomes, lysosomes) increased the release of CD63 and some endosomal and lysosomal markers, along with syntenin. These data suggest (but do not prove) that neutralization of lysosomes prevents their fusion with MVBs, allowing more MVBs to fuse with the plasma membrane than otherwise would. An orthogonal approach is using a mutant of CD63 that doesn't localize to lysosomes. This leads to localization to both the plasma membrane and small internal vesicular organelles and also enhances secretion of CD63 in EVs. This is taken as proof that more small EVs are released from HeLa cells as ectosomes than as exosomes – although it seems possible that CD63 itself affects biogenesis (this is actually shown) no matter where it is, but that many of the exosomes formed in MVBs are degraded in lysosomes. So that result may not be generalizable to all EVs formed under steady state. Overall, this is a very interesting paper that addresses an issue in the field that is quite important. That is, can small EVs be distinguished as ectosomes or exosomes based on protein cargo markers and what proportion of small EVs are in those categories?

As it was all done in one cell line, it is a bit difficult to be conclusive about what proportion of small EVs released from cells are exosomes versus ectosomes. However, the authors do convincingly show that CD63 is an excellent marker of exosomes whereas CD9 is much more likely to mark small ectosomes. There is much to like about this study, but in some places the data are quite thin that underlie the conclusions. In addition, some of the data may be over- or mis-interpreted. Specific comments are:

We thank this reviewer for his/her thorough evaluation of our work, and we have tried to strengthen our data and eliminate over- or mis-interpretations as detailed below.

1. There is missing statistical analysis for most of the figures and many of the experiments do not have adequate replicates (where noted). It is hard to see how one can make conclusions without enough replicates and statistical tests. All quantitations (e.g. the Western blot graphs) should have statistics and adequate replicates (e.g. cell stainings). In some cases, there is not a notation of how many independent replicates and/or cells/image fields there are. Typically for cell images one would want ~10-20 fields per biological replicate and 3 biological replicates.

Concerning statistics, we would like to highlight to this reviewer a very interesting article published in 2012 in Nature (Vaux DL (2012). Know when your numbers are significant. Nature 492 : 180-181), which clearly explains why statistics per se are not to be considered as the graal for experimental biologists, who cannot deal with very large number of samples. Instead, Vaux explains that “When N is only 2 or 3, it would be more transparent to just plot

the independent data points, and let the readers interpret the data for themselves, rather than showing possibly misleading P values or error bars and drawing statistical inferences. If the data in an experiment are equivocal, or the effect size is small, it is much better to come up with an extra, mechanistically different, experiment to test the hypothesis, than to repeat the same experiment until P is less than 0.05. “. Following this advice, we had performed most of our experiments as 3 independent biological replicates, and displayed the independent data points, without performing statistical tests, but instead considering that experiments showing similar trends for 3 biological replicates were providing reliable information. Some of the experiments had indeed been done only as 2 independent biological replicates (as clearly visible on the figures showing individual data points), when they were not providing a crucial, but rather a confirmatory, information.

We have now repeated these experiments to have at least 3 biological replicates, and performed additional replicates for the experiments showing some level of variability, but also when we had to generate new samples to perform new analyses (such as for Western blots of figure 7e). We thus now provide statistical tests for all experiments including 3 or more biological replicates, although, as explained above, we do not think that these tests allow better evaluation of our data.

A few experiments are still performed only twice independently:

1) imaging of the RUSH-CD81 construct (supp figure 5), which conveys an information that we thought was useful to share with the readers (that CD81 is most likely not an exosome marker), but which is not the core of the article.

2) the intracellular immuno-EM quantification (figure 2c), which has now been performed a second time independently and shows consistent results between the two independent experiments: a new quantification panel is shown in figure 2c.

3) Novel confocal analyses of colocalization of the tetraspanins and different intracellular markers, shown in figures 2d (CD63-Cherry, CD9-GFP and Rab7), figure 1e, and new suppl figure 3a (CD63-WT and -YA). Figure 1e is only illustrative of an initial observation confirmed by several other consistent experiments performed in the rest of the article, and we do not consider useful to quantify the level of CD9 and CD63 colocalization. For colocalization with markers of intracellular compartments (figure 2d and suppl figure 3a), we quantified 3 dozen cells in multiple fields of the 2 independent experiments, and observed clear-cut differences. Both EM quantitative analysis and confocal experiments involving multiple compartments stainings are very time-consuming, and provide results which confirm each other by different approaches, thus we hope that the reviewer and editor will accept our choice to follow the recommendations of Vaulx 2012 for these particular experiments.

Number of cells, fields and replicates, are now clearly indicated (at least 10 cells, up to 15, in at least 5 fields, up to 10, were analyzed in each independent experiment for quantifications) in the M&M paragraphs “Immunofluorescence on cells” and “Live Imaging of RUSH constructs”.

2. While replicates and image fields are mentioned in some figure legends, they are not in others (E.g. Supp Fig 5)

Replicates and numbers of image fields are now indicated in all figure legends.

3. Supp Fig 1. This is a very nice experiment, combining the conditioned media of KO cells. However, the interpretation does not seem totally correct as there is some pull-down of CD63 from the combined KO CM in the CD9 IP. Also, there is no quantitation of the results. The point about whether CD9 and CD63 are really present on the same EVs could be addressed by immunoelectron microscopy.

For the co-IP control experiment of supp figure 1 (now suppl figure 1a), we had performed it as a technical control, and thus did not consider it had to be quantified (like technical controls of antibody specificity or others). This experiment showed, when using mixed KO EVs, no CD9 pulled down by anti-CD63, and a very weak signal for CD63 pulled down by anti-CD9. Nonetheless, we have now repeated the IP experiment of mixed KO cell-derived EVs and show quantifications in suppl figure 1b. For technical reasons, we had to use a different source of anti-CD63 and anti-CD9 beads, and we show 2 novel experiments with these tools, where WT and mixed KO EVs were used side-by-side. Quantification of the signals shows around 50% decrease of co-isolation of the other TSPAN when mixed KO EVs than WT EVs are used. These results are described in the results section p.3, in the M&M section "Immunoisolation of EVs by anti-CD63 and -CD9" and in the legend of suppl fig1.

Finally, we have followed this reviewer suggestion, and we show a new panel d in figure 1, illustrating by immuno-EM the existence of CD9+, CD63+ and double CD9/CD63+ EVs in the 200K pellets of HeLa cells.

4. Supp Fig 3 and Fig 3, what are the small compartments that the CD63-YA-GFP goes to? Some labeling with early, late, and recycling endosome compartments are in order (e.g. Rab5, Rab7, Rab11...) as well as colocalization with lysosomal markers (LAMP1)

This is an interesting point, and we have now addressed this reviewers' question. We have performed new immunofluorescence labelling with antibodies to various intracellular compartments (RAB7, RAB5, EEA1 and LAMP1) in cells expressing CD63-WT-GFP or CD63-YA-GFP. By measuring co-localization of each of these markers with both CD63 molecules, we observed lower overall level of colocalization of CD63-YA than CD63-WT with all these intracellular compartments, consistent with its higher level of expression at the cell surface, and higher co-localization with LAMP1 of CD63-WT than CD63-YA, consistent with the absence of lysosome-targeting motif in the latter. Finally, CD63-YA retained stronger colocalization with EEA1 and RAB7 than with the other markers, suggesting that, when intracellular, and among the markers we tested, CD63-YA is both in EEA1-positive early endosomes and in RAB7-positive late endosomes. These results are now described p.5 and in new suppl figure 3a.

5. Supp Fig 4: It looks like CD63-YA-GFP and CD63-GFP expression both increase the total number of small EVs released from cells at 24 h in comparison to nontransfected cells and/or CD9-GFP cells. It seems like these data are in contradiction to the conclusion that more ectosomes are released from HeLa Cells than exosomes. It would seem to me that the number of EVs is a more reliable readout of EV biogenesis than CD63 levels. It also seems like all of the constructs are affecting the process they are being used to measure. How do the authors interpret these data and how do the levels of expressed proteins compare to each other? I don't see a Western blot directly comparing the levels of CD63-YA-GFP, CD63-GFP, and CD9-GFP in transfected cells to each other using a GFP antibody.

Supp Figure 4 indeed showed a clear increase of EV release by cells transfected with the RUSH constructs, CD63-WT, CD63-YA, but also CD9, as the reviewer could see from the 3 individual biological replicates displayed. To understand whether the tetraspanin themselves were influencing EV release, as suggested by this reviewer, we performed new experiments using 2 additional plasmids for transient transfection of HeLa cells before EV secretion: the empty backbone used to design the RUSH plasmids (pIRES-Neo), and a Str-KDEL RUSH plasmid encoding GFP fused to a GPI-anchor sequence, in addition to the RUSH plasmids CD63-GFP and CD9-GFP used in the study (see below figure for reviewer). We observed an equivalent increase of EV release for all these plasmids, as compared to EV release by untransfected cells, showing that the effect on EV secretion is due to the transfection itself (maybe linked to the slightly lower cell viability induced) rather than to CD63 or CD9 overexpression, or to the RUSH system.

Figure for reviewer: Cell viability and Number of particles released by cells 24h after transient transfection and biotin exposure, with CD9-GFP-RUSH and CD63-GFP-RUSH plasmids as compared to another RUSH (GFP-GPI) and the empty backbone of the RUSH plasmids (pIRES-Neo).

Since the RUSH and transient transfection system is used here to compare samples expressing the different constructs, and thus obtained in the same conditions of transient transfection, this increased EV release affects the samples in an equivalent manner and thus does not interfere with the interpretations. In addition, the question addressed here is the efficiency of secretion in EVs of the different types of TSPANs, therefore we need to quantify specifically the TSPAN-associated GFP signal, rather than the total number of EVs, which come both from transfected and untransfected cells in the culture.

Concerning the level of GFP expression in cells after transfection with the different RUSH plasmids, in all such experiments performed for subsequent EV analysis, the efficiency of transfection was systematically measured by flow cytometry: as previously indicated in the legend of figure 5B, the % of GFP+ cells was similar in different biological replicates, and independent of the sequence encoded by the RUSH plasmid. We have now generated a new suppl figure 4a to display these results and describe them in the results section p.6. We hope that the reviewer will agree that it is as informative as a Western blot (suggested by the reviewer).

6. Fig 5C: The band intensity for the GFP-YA-CD63 looks to be increased substantially by Bafilomycin, unlike what is stated in the text and what is shown in the bar graph. Why is that? Is it not a representative blot or was there a problem in quantitation of the blot? The Western blot previously shown in figure 5C corresponded to the CD63-YA point on the graph that was the most affected by BafA1. We performed a third independent replicate of this experiment, and replaced the WB by one showing a lower increase induced by BafA1.

7. Related to Fig 6 - Bottom of page 6 - This sentence is very confusing: "This suggests that a common mechanism of release (or subcellular origin) of CD63 EVs and a fraction of CD9 EVs and another mechanism specific of CD63 EVs are both equally important early after CD63 and CD9 synthesis." The sentence is convoluted and hard to understand. I also think that the authors may be reading too much into their data – it seems pretty clear that CD63 is mostly in endosomes and CD9 is mostly at the plasma membrane and the EVs reflect that. There is some overlap due to natural trafficking between those two compartments. Following this reviewer and reviewer 3's comment, we have deleted this paragraph of results, and do not try anymore to interpret the % of proteins identified as CD9- or CD63-EV associated. We are now showing the volcano plots of proteins quantified in the CD63 versus CD9-EVs (former suppl figure 4B) as figure 6a.

8. Fig 6B. The statistics that are done are a little hard to understand – the legend says it is in relation to the representation in the database – what does that mean with respect to this study? Does that mean that there are not comparisons within the study? For example, there are a lot of mitochondrial proteins in the 3h CD9+ sample but not at 24 h. Is that meaningful?

Indeed, as detailed in the results and legend of figure 6, the comparisons were performed only with the reference database, to identify the specific enrichment in EVs of proteins from a particular subcellular organelle. We thus did not comment on the higher apparent representation of Golgi and mitochondrial proteins in CD9-EVs at 3h as compared to 24h, because these % were not significantly higher than those of the whole cells (respectively 3,8% for Golgi and 13,1% for mitochondria), probably due to a low amount of proteins from these compartments in EVs. We can make conclusions of the comparison of the GO term enrichment between 3h and 24h when there is a significant enrichment in one compartment in a condition and not in the other one, but it is true that when there are significant enrichments or no enrichments in both conditions it is difficult to say whether the possible difference of percentage of enrichment is meaningful or not.

We have reformulated the description of % and p-values in the legend of figure 6b, we hope that it is now easier to understand.

9. Fig 6 (potential Discussion points) – could the authors comment on the relationship of the Endosome and Lysosome categories to Late Endosomal proteins, such as would be expected to represent MVBs? Also, how does the 24 h sample relate to steady state samples?

The manually curated database of intracellular proteins generated by G. Borner (Itzhak, eLife 2016) separates lysosomal proteins (mainly lysosomal enzymes and a few lysosome-specific

transmembrane proteins, that can be classified MVB associated in other databases like Panther) from all other proteins found in endosomal compartments, whatever their stage of the endocytic process (early, recycling, and late endosomes including MVBs). We have included this clarification in the last paragraph of the 'Proteomic analysis' results section p.8 Following this reviewer's and reviewer 1's question about the time points of EV analysis for proteomic studies, we have performed new quantification of CD9 and CD63 colocalization at different time-points after biotin addition, shown in new suppl figure 4e. The 24h time point shows a slightly lower colocalization level than the 3h time point, and a very similar comparable image as the endogenous distribution shown in figure 1e, and we thus consider it representative of the steady-state distribution.

10. Discussion – there are some confusing sentences, like: "Concerning the PM-EV markers: SLC3A2 is the heavy chain of various heterodimeric amino-acid transporters, and one of its possible partners, SLC7A5 (also named E16 or LAT-1)23, was less specific of the CD9-EVs, but not enriched in the CD63-EVs, thus confirming an ectosomal release."

Thank you for pointing at an unclear sentence: we have simplified this sentence to highlight the simultaneous presence of the two partners in CD9-enriched EVs, hence ectosomes, at 24h.

11. It is a bit surprising that some EV characterization, as outlined in MISEV2018 and the earlier MISEV, are missing from this study – such as TEM and checking negative markers on Western blot.

Reviewer 1 made the same comment: please see our detailed answer to Reviewer 1's comment 1 above. In this revised version, we provide new panels in figures 1 (control HeLa EVs) and 7 (EVs released upon BafA1 and GW4869 treatment): Western blot panels showing additional markers for cytosolic proteins (Syntenin1) and two types of negative controls (calnexin and AChE), additional NTA-based particle quantification and sizing, and new EM-based quantification and sizing. We have also submitted our experimental conditions to the EVTRACK database. Reviewers and editor may access and check the submission of experimental parameters to the EV-TRACK knowledgebase via the following URL: <http://evtrack.org/review.php>. Please use the EV-TRACK ID (EV210105) and the last name of the first author (Mathieu) to access our submission

Reviewer #3 (Remarks to the Author):

Technical review of the used LC-MS of captured extracellular vesicles.

Matthieu et al. describe the use of an elegant system making use of the RUSH system in which they 'stall' release of their proteins of interest from e.g. the ER to image the synchronized trafficking of proteins used as specific EV markers. The study is well described and addresses an important question in the field of EV biogenesis. The aim of this review is

to assess the technical quality of the performed MS-based proteomics experiments to identify EV subtype-specific proteomes.

In their approach the authors used their adapted RUSH system to immunocapture CD63- and CD9-eGFP EVs, using an anti-GFP antibody. This is an elegant combination that allows for the enrichment of freshly secreted EVs using the same antibody for both conditions, with a non-transfected control. The authors performed all experiments 4-fold and performed the statistical analysis with FDR correction. Overall the data is robust and well presented. A few small comments below.

We thank this reviewer for his/her positive evaluation of our work.

1. The authors make some claims on the end of page 6 and start of page 7, based on the observation of a slight increase of common (CD63/CD9) proteins (44% to 48%) and a decrease of CD63 specific proteins (37% to 32%) from 3hrs to 24hrs. They state this observation indicates different mechanisms of release for the different populations and increased dominance of one over time. Given the very small differences, it's questionable these observations are statistically significant and importantly, an observation in a single cell line (Hela) is not sufficient for such general statements. If the authors really believe this, they should show evidence for this in multiple cell lines and perform the proper statistical analysis.

Based on this reviewer and reviewer 2's comment, we agree that this interpretation of the proteomic data was a bit complicated and not so crucial, we have thus decided to delete this paragraph from the text and to replace former figure 6A (pie-chart of % of proteins in each category) by the volcano plots of former supp figure 4B, now figure 6a.

2. Supp figure 3B is very busy and difficult to interpret at first glance. The quality of the representation can be improved.

We assume that the reviewer meant former supp figure 4B: volcano plots of the proteomic analysis. We have tried to simplify this figure by deleting most names of proteins (except those selected for further analysis in figure 7), and the figure replaces former figure 6A. We hope this makes it easier to read.

3. In the description of the MS analysis on page 16 one sentence is a bit poor: "The top 20 intense ions were subjected to Orbitrap for further fragmentation via high energy collision dissociation (HCD) activation" Ions are not fragmented in the Orbitrap, but used for readout of MS fragments from the HCD collision cell. "and a resolution of 15 000" the authors probably mean these fragments are analyzed at 15 000 resolution. " 1.3×10^5 " should be 10^5 .

Thank you for this suggestion, we have changed the corresponding sentence (now page 17).

REVIEWERS' COMMENTS

Reviewer #1 (Remarks to the Author):

The authors have adequately addressed the comments, by adding new experiments to further support the results. This is an exciting study contributing novel insights on exosomes versus ectosomes as well as the performance and suitability of inhibitors used in the field.

Reviewer #2 (Remarks to the Author):

Overall, the authors have done a great job at addressing my comments. Only one minor suggestion to improve the text:

On page 6, the sentence "In contrast, only minor increase in EVs was observed for CD63-YA-eGFP or CD9-eGFP upon BafA1 treatment (figure 5c), while the total amount of released EVs was increased for all cells (supp figure 4d)." is confusing.

I suggest rewording: "In contrast, only minor increases in CD63-YA-eGFP or CD9-eGFP CONTENT OF EVs WERE observed for upon BafA1 treatment (figure 5c), while the total amount of released EVs was increased for all cells (supp figure 4d)."

Reviewer #3 (Remarks to the Author):

The authors addressed all comments and the paper can be published

21/05/2021
NCOMMS-20-38099A

Point by point response to reviewers comments

REVIEWERS' COMMENTS

Reviewer #1 (Remarks to the Author):

The authors have adequately addressed the comments, by adding new experiments to further support the results. This is an exciting study contributing novel insights on exosomes versus ectosomes as well as the performance and suitability of inhibitors used in the field.

Thank you

Reviewer #2 (Remarks to the Author):

Overall, the authors have done a great job at addressing my comments. Only one minor suggestion to improve the text:

On page 6, the sentence “In contrast, only minor increase in EVs was observed for CD63-YA-eGFP or CD9-eGFP upon BafA1 treatment (figure 5c), while the total amount of released EVs was increased for all cells (supp figure 4d).” is confusing.

I suggest rewording: “In contrast, only minor increases in CD63-YA-eGFP or CD9-eGFP CONTENT OF EVs WERE observed for upon BafA1 treatment (figure 5c), while the total amount of released EVs was increased for all cells (supp figure 4d).”

Thank you for the suggestion, we changed the sentence accordingly.

Reviewer #3 (Remarks to the Author):

The authors addressed all comments and the paper can be published

Thank you